# ProtInvTree: Deliberate Protein Inverse Folding with Reward-guided Tree Search

**Mengdi Liu[1,2], Xiaoxue Cheng[4], Zhangyang Gao[3], Hong Chang[1,2]\***,
**Cheng Tan[3], Shiguang Shan[1,2], Xilin Chen[1,2]**

[1] State Key Laboratory of AI Safety, Institute of Computing Technology, CAS, China
[2] University of Chinese Academy of Sciences (CAS), China
[3] AI Lab, Research Center for Industries of the Future, Westlake University
[4] Gaoling School of Artificial Intelligence, Renmin University of China
`{liumengdi23z, changhong, sgshan, xlchen}@ict.ac.cn`,
`chengxiaoxue@ruc.edu.cn, {gaozhangyang, tancheng}@westlake.edu.cn`

## Abstract

Designing protein sequences that fold into a target 3D structure—known as protein inverse folding—is a fundamental challenge in protein engineering. While recent deep learning methods have achieved impressive performance by recovering native sequences, they often overlook the one-to-many nature of the problem: multiple diverse sequences can fold into the same structure. This motivates the need for a generative model capable of designing diverse sequences while preserving structural consistency. To address this trade-off, we introduce **ProtInvTree**, the first reward-guided tree-search framework for protein inverse folding. ProtInvTree reformulates sequence generation as a *deliberate, step-wise decision-making process*, enabling the exploration of multiple design paths and exploitation of promising candidates through self-evaluation, lookahead, and backtracking. We propose a two-stage *focus-and-grounding* action mechanism that decouples position selection and residue generation. To efficiently evaluate intermediate states, we introduce a *jumpy denoising* strategy that avoids full rollouts. Built upon pretrained protein language models, ProtInvTree supports flexible test-time scaling by expanding the search depth and breadth without retraining. Empirically, ProtInvTree outperforms state-of-the-art baselines across multiple benchmarks, generating structurally consistent yet diverse sequences, including those far from the native ground truth. The code is available at `https://github.com/A4Bio/ProteinInvBench/`.

## 1 Introduction

Proteins are 3D folded linear chains of amino acids that perform essential biological functions, such as metabolic control, transmitting signals, and regulating cellular processes [21, 5]. Designing sequences of amino acids that fold into a desired protein structure, also known as protein "inverse folding" (IF) [58], is a crucial task with great potential for protein engineering and synthetic biology [27, 62, 8]. Recent deep learning approaches typically recover the native sequence conditioned on a target structure through the following three paradigms: *autoregressive generation*, which models sequence dependencies step-by-step [22, 23, 44, 20, 10]; *one-shot prediction*, which directly maps structure to sequence in a single forward operation [13, 32, 15]; and *iterative refinement*, which progressively improves an initial design through multiple passes [61, 11, 12, 63]. Despite of achieving impressive recovery performance, these methods often overlooked the inherently *one-to-many* nature of the problem [38, 33, 16], where multiple distinct amino acid sequences are capable of folding into the

---

\*Corresponding author.

39th Conference on Neural Information Processing Systems (NeurIPS 2025).

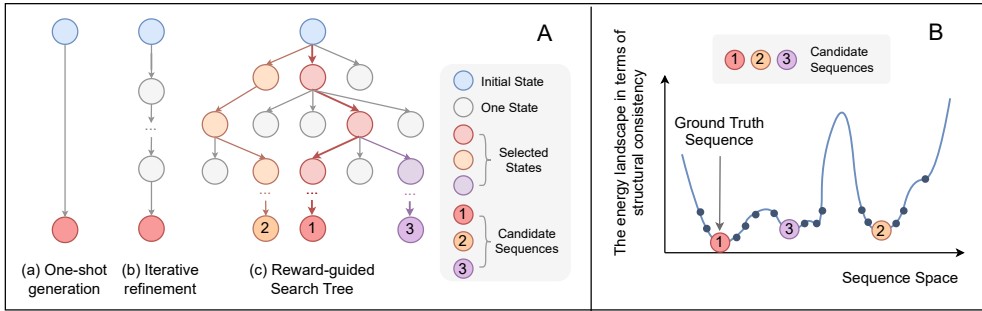

Figure 1: **(A)** Schematic illustration on various approaches of structure-based protein design. (a) One-shot generation that directly predicts full sequences from structure in one step, (b) Iterative refinement that first generates a full sequence and then improves it through multiple steps, and (c) our method that applies tree search to perform deliberate design. Each node denotes an intermediate sequence conditioned on the target structure, progressively expanding toward full generation. See the complete framework and details in Figure 2. **(B)** Energy landscape $E = e^{-\text{scTMScore}}$ over the sequence space with respect to the target structure, where the ground-truth sequence is marked and candidate sequences cluster around several local optima.

same protein backbone structure. As a result, rather than predicting a single native sequence, it is often desirable to generate a diverse set of sequences while preserving structural consistency.

This goal, however, reveals an inherent trade-off: while *diversity* prefers broad exploration of the sequence space, *structural consistency* strictly requires a feasible subspace that has good local residue compatibility and global foldability. To address this challenge, we advocate for a deliberate, step-wise design process that progressively explores the solution space of feasible sequences. Inspired by the dual process theory in cognitive science [25, 43, 40], where System 1 is characterized by fast, automatic, and heuristic-driven responses, while System 2 involves slow, deliberate, and analytical reasoning, we propose to model the protein design process as a *deliberate and iterative* decision-making process that should (1) explore multiple alternatives at each design step rather than one single candidate, (2) dynamically assess and revise each design step through lookahead and backtracking to optimize the overall designed sequence, and (3) maintain a structured decision history to support effective credit assignment and multi-step planning.

To this end, we propose **ProtInvTree**, a training-free framework for structure-based protein design that formalizes the design process as a sequence of branching decisions and leverages Monte Carlo Tree Search (MCTS) during generation. Specifically, we iteratively perform the design process, sampling multiple decisions at each step and looking ahead to compute reward signals that evaluate the quality of current choices, thereby guiding the overall sequence design. At each decision step, we introduce a two-stage focus-and-grounding action mechanism that first selects the positions in the sequence to modify (*focus*) and then generates new residues at these positions (*grounding*). Moreover, we employ fast, jumpy denoising as an evaluation mechanism, efficiently estimating trajectory quality without costly forward model rollouts. Through these designs, ProtInvTree is capable of making globally optimal decisions rather than settling for locally optimal ones, which allows it to design novel yet plausible sequences that may deviate significantly from the native sequence (as shown in Fig. 1). Additionally, the explicit exploration of design paths may offer potential insights into the interpretability of protein sequences. To our knowledge, this is the first work to apply a tree-search framework to structure-based protein sequence design.

Empirically, we comprehensively evaluate ProtInvTree across fixed-backbone and de novo protein design tasks. We demonstrate that ProtInvTree outperforms state-of-the-art baselines and excels in the design of plausible proteins with high structural consistency. Besides, it achieves Pareto-optimal trade-offs in both the scTMscore-diversity and scTMscore-novelty. Notably, we observe that existing approaches aggressively optimizing for sequence recovery achieve limited novelty at the same scTMscore level. Further analyses reveal that increasing planning depth and expansion width can effectively improve structural consistency, demonstrating that the paradigm of test-time scaling can effectively unlock the potential of pretrained protein language models (PLMs).

In summary, our **contributions** are as follows:

- We propose **ProtInvTree**, the first test-time reward-guided tree search framework for protein inverse folding. It formulates protein design as a deliberate, step-wise decision process, enabling *exploration* of multiple trajectories and *exploitation* of promising candidates.

- We introduce a two-stage **focus-and-grounding** mechanism decoupling position selection and residue generation and a **fast, jumpy denoising** strategy for efficient reward evaluation.

- We demonstrate that ProtInvTree achieves state-of-the-art performance across multiple benchmarks, with a **test-time scaling capability** that improves both structural consistency and sequence diversity without retraining or fine-tuning.

## 2 Related Works

### 2.1 Protein Inverse Folding

Recently, AI algorithms have spurred a major revolution in modeling protein inverse folding, enabling accurate sequence design conditioned on target structures. Existing approaches can be broadly categorized into the following three paradigms based on their generation strategies.

**Autoregressive models** generate sequences residue-by-residue, conditioned on both the 3D structure and previously generated tokens. Pioneering models like GraphTrans [22] and GVP [23] introduced SE(3)-invariant graph encoders with attention or geometric modules. Later, models such as GCA [44], ESM-IF [20], and ProteinMPNN [10] incorporated global context and fine-grained pairwise distance modeling. These models offer accurate recovery but suffer from slow inference on long sequences.

**One-shot models** bypass iterative steps by directly predicting full sequences from structure. Pi-Fold [13] introduced an efficient graph encoder with an MLP decoder, achieving significant speedups and improved accuracy on long proteins. Uni-IF [15] generalizes this to multiple molecule types. These models are highly efficient but face challenges in maintaining global structural consistency.

**Iterative refinement methods** address this by first generating a full sequence and then improving it through multiple steps. AlphaDesign [12] and LMDesign [61] use confidence-aware predictors and pretrained sequence models for guided refinement. KWDesign [11] combines sequence and structure pretraining with an uncertainty-aware update mechanism. Recent works such as BridegIF [63] and GraDe-IF [57] apply diffusion to enhance diversity and structural compatibility. Fast non-autoregressive diffusion models like PMPnnDiff [54] accelerate inference while preserving accuracy.

Despite notable advances in protein inverse folding, most efforts focus on training-time improvements, while the inference phase remains underexplored. As large protein foundation models emerge, harnessing test-time computation to boost sequence quality and diversity becomes crucial for incentivizing their full potentials. To this end, we propose a novel paradigm based on tree-structured generation, which departs fundamentally from the three existing categories of approaches.

### 2.2 Test-time Scaling and MCTS

Test-time scaling refers to increasing computational resources during inference to enhance model output without modifying its parameters. This approach has gained significant attention in the field of large language models (LLMs), where performance is improved by generating multiple samples and using reward models for best-solution selection [41, 53, 4]. Various test-time search methods have been proposed [26, 46], including random sampling [49], self-consistency, and tree-search methods [56, 17, 59, 36]. Among them, MCTS, a heuristic search algorithm designed for decision-making tasks [55, 50, 7, 9, 39], has emerged as a powerful technique for structured exploration in the output space of large language models. It enables deliberate reasoning by simulating multiple generation trajectories, selectively expanding promising paths, and integrating reward feedback to guide inference toward high-quality outputs. Inspired by these advances, we are the first to extend the paradigm of test-time scaling to PLMs. Our proposed framework, ProtInvTree, leverages reward-guided tree search to perform deliberate, step-wise protein sequence generation, enabling test-time scaling for improved structural consistency and diversity.

## 3 Preliminaries

**Problem Definition.** The protein inverse folding problem seeks to determine the amino acid sequence $\mathbf{x}$ that folds into a given target structure $\mathbf{c}$. Here, $\mathbf{x} = [x_1, x_2, \ldots, x_L]$ represents the sequence of $L$ residues, where $x_i \in \{1, 2, \ldots, 20\}$ denotes the type of the $i$-th residue. The structure $\mathbf{c} = [c_1, c_2, \ldots, c_n] \in \mathbb{R}^{n \times 4 \times 3}$ specifies the Cartesian coordinates of the backbone atoms (N, C-$\alpha$, C, and optionally O) for each residue $\mathbf{c}_i$. The inverse folding problem can be formally expressed as:

$$f_\theta : \mathbf{c} \to \mathbf{x}, \tag{1}$$

where $\theta$ is the learnable parameter. Given that homologous proteins invariably exhibit similar structures, the solution for a given structure is not unique [16]. Hence, an ideal model should be capable of learning the underlying mapping from protein backbone structures to their corresponding sequence distributions $p_\theta(\mathbf{x}|\mathbf{c})$.

**Iterative Denoising.** Recently, diffusion models [19, 42, 37, 35, 30] have demonstrated remarkable capabilities in the field of life science [1, 2, 47, 51, 48, 31] and have achieved notable success in protein inverse folding [12, 61, 11, 63]. These methods, including iterative refinement models, formulate the task as an iterative denoising process that refines the sequence step by step. Following this paradigm, we adopt this strategy to progressively construct the sequence. Formally, starting from an initially corrupted sequence $\mathbf{x}_0$, the model iteratively denoises the sequence into a complete design $\mathbf{x}_T$ through a series of conditional reverse transitions, where $\mathbf{x}_0$ and $\mathbf{x}_T$ differ from the definition in diffusion modeling:

$$p_\theta(\mathbf{x}_T \mid \mathbf{x}_0, \mathbf{c}) = \prod_{t=1}^{T} p_\theta(\mathbf{x}_{t+1} \mid \mathbf{x}_t, \mathbf{c}), \tag{2}$$

where $\mathbf{x}_t$ represents the intermediate sequence at step $t$, with a subset of amino acids remaining unfilled (e.g., represented by [MASK] tokens), and $\mathbf{c}$ denotes the target backbone structure. Each reverse step $p_\theta(\mathbf{x}_{t+1} \mid \mathbf{x}_t, \mathbf{c})$ refines the current sequence while preserving the structural context.

## 4 ProtInvTree: Deliberate Protein Inverse Folding Framework

In this section, we propose a reward-guided tree-search framework for deliberate protein inverse folding. We first formulate the iterative denoising as a tree-based Markov decision process (MDP), enabling structured exploration over multiple trajectories (Section 4.1). Then we describe the MCTS procedure employed to identify diverse and high-quality sequences that are consistent with the target backbone structure (Section 4.2). Finally, we introduce two designs of the action and reward components (Sections 4.3 and 4.4), which define how the sequence is updated at each step and how intermediate states are evaluated during the search process. We present the detailed algorithm to formalize the entire framework in the appendix.

### 4.1 Tree-based MDP Formulation

As described in Section 3, while the step-wise denoising process is effective, it lacks the ability to incorporate intermediate feedback, track uncertainty, and revise previous decisions. To overcome these limitations, we reformulate the iterative denoising process as a *tree-based Markov decision process* for structured, feedback-aware generation. In this tree structure, each **node** represents a state $\mathbf{s}_t$, each **branch** corresponds to an action $\mathbf{a}_t$, and each node is assigned a **value** that reflects the reward $r_t$ at that state. Specifically, we define the concepts in the tree search framework as follows:

$$\mathbf{s}_t \triangleq (\mathbf{c}, \mathbf{x}_t), \qquad \mathbf{a}_t \triangleq \{(i_k, x_{i_k})\}_{k=1}^{K_t}, \qquad r_t \triangleq R(\mathbf{s}_t, \mathbf{a}_t), \qquad \pi(\mathbf{a}_t \mid \mathbf{s}_t) \triangleq p_\theta(\mathbf{x}_{t+1} \mid \mathbf{x}_t, \mathbf{c}).$$

Here, the state $\mathbf{s}_t$ consists of the target backbone structure $\mathbf{c}$ and a partially generated sequence $\mathbf{x}_t$. The action $\mathbf{a}_t$ corresponds to the selection of positions in the sequence and modification of new residues, as detailed in Section 4.3. The reward $r_t$ is computed by a reward function $R(\mathbf{s}_t, \mathbf{a}_t)$, which evaluates the structural consistency of the modified sequence, as described in Section 4.4. The policy model $\pi(\mathbf{a}_t \mid \mathbf{s}_t)$ generates the next partial sequence $\mathbf{x}_{t+1}$ based on the current state $\mathbf{s}_t$. It is parameterized by a structurally modulated protein language model (PLM). A trajectory in the multi-step Markov decision process is defined as a sequence of state-action-reward transitions:

$$\tau = [(\mathbf{s}_0, \mathbf{a}_0, r_0), (\mathbf{s}_1, \mathbf{a}_1, r_1), \ldots, (\mathbf{s}_T, \mathbf{a}_T, r_T)],$$

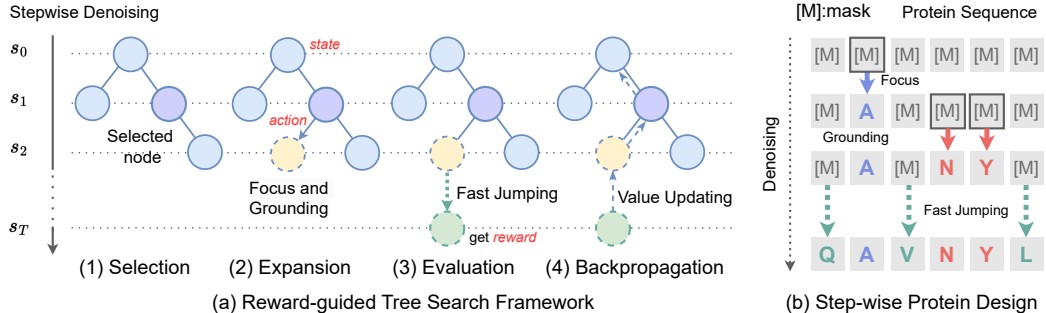

Figure 2: The framework of ProtInvTree. (a) The four steps of reward-guided tree search—*Selection, Expansion, Evaluation, and Backpropagation*—are illustrated on a partial denoising tree. Each node corresponds to a partially denoised subsequence. After a new node is expanded, "jumpy" denoising is performed to quickly estimate its value, which is then backpropagated along the path in the tree. (b) Illustration of how a sequence is generated step by step. Masked tokens in the sequence are progressively infilling through a focus-and-grounding mechanism.

where each transition corresponds to an incremental update of the sequence. By reformulating the sequence design process from a linear chain into a tree structure, our framework enables deliberate planning over multiple generation trajectories, facilitates the incorporation of intermediate feedback from structural evaluations, and supports systematic revision of prior design decisions.

## 4.2 Reward-guided Tree Search

In our approach, the reward-guided tree search process operates as an iterative procedure. As illustrated in Figure 2, it comprises four key steps: selection, expansion, evaluation, and backpropagation. The details of each step are described as below.

**Selection.** The selection process begins at the root node $s_0$ and identifies the leaf node with the highest exploration potential, determined by the UCT (Upper Confidence Bounds applied to Trees) [28] score. The UCT score is computed as follows:

$$UCT(\mathbf{s}_t) = V(\mathbf{s}_t) + w\sqrt{\frac{\ln N(\mathbf{p})}{N(\mathbf{s}_t)}}, \tag{3}$$

where $w$ is a hyperparameter that balances exploitation (i.e., node value $V(\mathbf{s}_t)$) and exploration (i.e., visit count $N(\mathbf{s}_t)$), and $\mathbf{p}$ denotes the parent node of $\mathbf{s}_t$.

**Expansion.** After selecting the node with the highest UCT score, it is expanded by generating multiple child nodes. Conditioned on the current state $\mathbf{s}_t$, which consists of the target structure $\mathbf{c}$ and current sequence $\mathbf{x}_t$, the policy model samples $K$ candidate sequences for the next step:

$$\{\mathbf{x}_{t+1}^{(k)}\}_{k=1}^K \sim \pi_\theta(\mathbf{a}_t \mid \mathbf{s}_t) \triangleq p_\theta(\mathbf{x}_{t+1} \mid \mathbf{x}_t, \mathbf{c}). \tag{4}$$

Each candidate sequence $\mathbf{x}_{t+1}^{(k)}$ constitutes a new child state $\mathbf{s}_{t+1}^{(k)} = (\mathbf{c}, \mathbf{x}_{t+1}^{(k)})$, which is added to the search tree as an expansion of the selected node. The details of the candidate construction process by policy model are described in Section 4.3.

**Evaluation.** Each expanded node is evaluated to determine its value $V(\mathbf{s}_{t+1})$. As described in Equation 12, we first perform rollouts that complete the state $\mathbf{s}_{t+1}$ via sampling $m$ fully generated sequences, and then assess them with a reward model, assigning the average reward $r_{t+1}$ as the node value $V(\mathbf{s}_{t+1})$. The details of the reward function and evaluation process are provided in Section 4.4.

**Backpropagation.** After evaluating the expanded nodes, their values are backpropagated along the traversal path to update the visit counts and value scores of the ancestor nodes $\mathbf{s}_j$ $(0 \le j \le t)$. The updates are performed using the following equations:

$$N_{\text{new}}(\mathbf{s}_j) = N_{\text{old}}(\mathbf{s}_j) + 1, \tag{5}$$

$$V_{\text{new}}(\mathbf{s}_j) = \frac{V_{\text{old}}(\mathbf{s}_j)N_{\text{old}}(\mathbf{s}_j) + r_{t+1}}{N_{\text{new}}(\mathbf{s}_j)}, \tag{6}$$

where $N_{\text{old}}(\mathbf{s}_j)$ and $V_{\text{old}}(\mathbf{s}_j)$ represent the previous visit count and value score of node $\mathbf{s}_j$, respectively, and $r_{t+1}$ is the reward obtained during the evaluation step.

The four stages described above are performed iteratively until the terminal state is reached. We define two termination conditions for MCTS as follows: (1) The maximum number of MCTS iterations, $M$, is reached. (2) A terminal node is encountered with a reward exceeding a predefined threshold, indicating strong structural consistency and high-quality design. Once the tree search is complete, the optimal path is selected greedily by prioritizing nodes with the highest scores.

### 4.3 Focus-and-Grounding Action

To generate candidate transitions from each intermediate state $\mathbf{s}_t$, we propose a two-stage *Focus-and-Grounding* action mechanism (see illustration in Fig. 2b). At each denoising step, the model explicitly decomposes the sequence updating process into identifying **where** to modify (Focus) and determining **what** token to generate at the selected position (Grounding).

Formally, the *Focus* operation $\mathcal{F}(\cdot)$ defines a position selection distribution $p_\phi(i \mid \mathbf{s}_t)$ over all positions, from which the top-$K_t$ positions with the highest probabilities are selected:

$$\mathcal{F}(\mathbf{s}_t) = \text{argsort}_{i \in \{1,\dots,L\}} \left( p_\phi(i \mid \mathbf{s}_t), K_t \right), \tag{7}$$

where $L$ denotes the sequence length and $i$ indicates the targeted position for refinement. Conditioned on the focused positions $\{i_1, \dots, i_N\}$, the *Grounding* operation defines a distribution over amino acid types, specifying the generated token:

$$\mathcal{G}(\mathbf{s}_t, i) = p_\psi(x_i \mid \mathbf{s}_t, i), \ i \in \mathcal{F}(\mathbf{s}_t), \tag{8}$$

where each $p_\psi(x_i \mid \mathbf{s}_t, i)$ predicts the residue $x_i \in \mathcal{V}$ for position $i$, and $\mathcal{V}$ denotes the amino acid vocabulary. The overall policy is factorized as the product of the Focus and Grounding distributions:

$$\pi_\theta(\mathbf{a}_t \mid \mathbf{s}_t) = \prod_{i \in \mathcal{F}(\mathbf{s}_t)} p_\phi(i \mid \mathbf{s}_t) \cdot p_\psi(x_i \mid \mathbf{s}_t, i), \tag{9}$$

In practice, the selected position set $\{i_1, \dots, i_N\}$ is a random subset of sequence positions (more selection strategies comparison is provided in appendix), and each token $x_i$ is generated by a structurally modulated PLM conditioned on the backbone structure $\mathbf{c}$ and the partial sequence context. This two-stage action design enables precise localization of modifications, ensuring structural coherence and enhancing search efficiency throughout the generation process.

### 4.4 Jumpy Denoising for Fast Reward

In the MCTS procedure, evaluating a node far from a leaf node is challenging, as the intermediate nodes are not fully expanded. This is typically addressed in one of two ways: employing forward dynamics models to simulate complete trajectories, which is computationally expensive, or approximating node values via bootstrapping methods, which are faster but less accurate. Effectively integrating these evaluation strategies into ProtInvTree remains an open challenge.

To address this, we introduce a *Jumpy Denoising* strategy to accelerate the evaluation process, which is a rapid, single-step DDIM-based [42] sampling process:

$$\tilde{\mathbf{x}}_T \sim \mathcal{J}(\mathbf{x}_{t+1}, \mathbf{c}), \tag{10}$$

where $\mathcal{J}(\cdot)$ approximates the reverse denoising distribution $p(\mathbf{x}_T \mid \mathbf{x}_{t+1}, \mathbf{c})$. Here, $\mathbf{x}_{t+1}$ is obtained through action $\mathbf{a}_t$ at step $t$. We define the reward function $R(\mathbf{s}_t, \mathbf{a}_t)$ as the structural consistency feedback obtained by comparing the folding results from the sampled sequence $\tilde{\mathbf{x}}_T$ and the input structure $\mathbf{c}$, formulated as:

$$R(\mathbf{s}_t, \mathbf{a}_t) = \text{TMScore}(f(\tilde{\mathbf{x}}_T), \mathbf{c}), \tag{11}$$

where $f$ is the protein folding algorithm. $\text{TMScore}(\cdot, \cdot)$ is a widely used metric for measuring protein structure similarity. This jumpy denoising strategy significantly reduces computational overhead while maintaining a reliable approximation of the final reward.

Table 1: Structure consistency performance comparison between ProtInvTree and different baseline approaches on the CATH 4.2 dataset. The split of "Short", "Single-chain" and "All" is the same as previous works. The **best** and second-best results are labeled with bold and underline.

| Models | Trainable/Total Params. | scT-Mscore (↑) | | | RMSD (↓) | | |
|---|---|---|---|---|---|---|---|
| | | Short | Single-chain | All | Short | Single-chain | All |
| StructGNN [22] | 1.4M/1.4M | 0.616 | 0.646 | 0.751 | 2.439 | 2.702 | 2.327 |
| GraphTrans [22] | 1.5M/1.5M | 0.590 | 0.635 | 0.744 | 2.356 | 2.678 | 2.351 |
| GCA [45] | 2.1M/2.1M | 0.606 | 0.646 | 0.755 | 2.430 | 2.596 | 2.226 |
| GVP [24] | 0.9M/0.9M | 0.611 | 0.662 | 0.771 | 2.289 | 2.542 | 2.181 |
| ProteinMPNN [10] | 1.9M/1.9M | 0.636 | 0.692 | 0.795 | 2.310 | 2.370 | 2.009 |
| AlphaDesign [12] | 3.6M/3.6M | 0.646 | 0.693 | 0.814 | 2.271 | 2.422 | 1.969 |
| PiFold [13] | 5.8M/5.8M | 0.655 | 0.700 | 0.842 | 2.203 | 2.355 | 1.723 |
| UniIF [15] | 5.4M/5.4M | 0.660 | 0.709 | 0.845 | 2.168 | 2.298 | 1.680 |
| LM-Design (ESM-1b) [61] | 6.9M/650M | 0.663 | 0.714 | 0.849 | 2.150 | 2.240 | 1.638 |
| KW-Design (ESM-2) [11] | 54.49M/650M | 0.676 | 0.729 | 0.858 | 2.101 | 2.148 | 1.566 |
| ESM-3 [18] | 1.4B/1.4B | 0.668 | 0.692 | 0.816 | 2.060 | 2.387 | 2.135 |
| **ProtInvTree** (ESM-3) | 0M/1.4B | **0.768** | **0.800** | **0.881** | **1.902** | **2.136** | **1.513** |

Table 2: Structural consistency comparison between ProtInvTree and baseline approaches on CATH 4.3 datasets. The **best** and second-best results are labeled with bold and underline.

| Model | sc-TMscore (↑) | RMSD (↓) |
|---|---|---|
| StructGNN [22] | 0.693 | 2.563 |
| GraphTrans [22] | 0.690 | 2.614 |
| GCA [45] | 0.698 | 2.525 |
| GVP [24] | 0.713 | 2.509 |
| ProteinMPNN [10] | 0.743 | 2.238 |
| AlphaDesign [12] | 0.749 | 2.230 |
| PiFold [13] | 0.785 | 1.949 |
| KW-Design (ESM-2) [11] | 0.818 | 1.751 |
| ESM-3 [18] | 0.775 | 2.074 |
| ProtInvTree (ESM-3) | **0.835** | **1.702** |

Table 3: De novo protein design results on TS45 datasets. We compare structural consistency of the following methods. The **best** and second-best results are labeled with bold and underline.

| Model | sc-TMscore (↑) | RMSD (↓) |
|---|---|---|
| StructGNN [22] | 0.631 | 3.336 |
| GraphTrans [22] | 0.618 | 3.276 |
| GCA [45] | 0.660 | 3.226 |
| GVP [24] | 0.652 | 3.245 |
| ProteinMPNN [10] | 0.668 | 3.142 |
| AlphaDesign [12] | 0.660 | 3.167 |
| PiFold [13] | 0.699 | 2.875 |
| KWDesign (ESM-2) [11] | 0.711 | 2.643 |
| ESM-3 [18] | 0.690 | 2.958 |
| ProteinInvTree (ESM-3) | **0.724** | **2.513** |

# 5 Experiments

## 5.1 Experimental Setup

**Datasets.** We conduct experiments on both **CATH v4.2** and **CATH v4.3** [34], where proteins are categorized based on the CATH hierarchical classification of protein structure, to ensure a comprehensive analysis. Following the standard data splitting [22, 20], CATH v4.2 dataset consists of 18,024 proteins for training, 608 proteins for validation, and 1,120 proteins for testing; CATH v4.3 dataset consists of 16,153 proteins for training, 1,457 proteins for validation, and 1,797 proteins for testing. We also include a set of de novo proteins collected from the CASP15 competition to provide a more realistic assessment. Following the previous work ProtInvBench [14], we download the public TS-domains structures from CASP15 which consists of 45 structures, namely **TS45**.

**Evaluation Metrics.** For evaluation metrics, we use **sc-TMscore** [60] and **RMSD** [6] to evaluate structural consistency. We define **diversity** as the average proportion of differing residues across all pairs of generated sequences and define **novelty** as $1 - $ **recovery**. Details of all metrics are provided in the appendix. Following previous studies [22, 20], we report them on three settings, namely short proteins (length $\leq 100$), single-chain proteins (labeled with 1 chain in CATH), and all proteins.

**Baselines.** We compare ProtInvTree with several state-of-the-art baselines, categorized into three groups: (1) autoregressive models, including StructGNN [22], GraphTrans [22], GCA [45], GVP [24], and ProteinMPNN [10]; (2) the one-shot model, PiFold [13], UniIF [15]; (3) iterative models, including AlphaDesign [12], LM-Design [61], KW-Design [11].

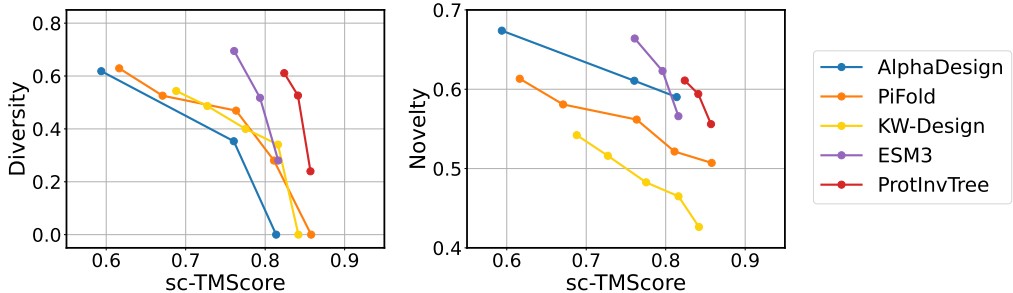

Figure 3: Pareto comparison of structural consistency (sc-TMScore) against diversity (left) and novelty (right) across different protein sequence design methods. Each curve represents a specific method evaluated under different sampling temperatures.

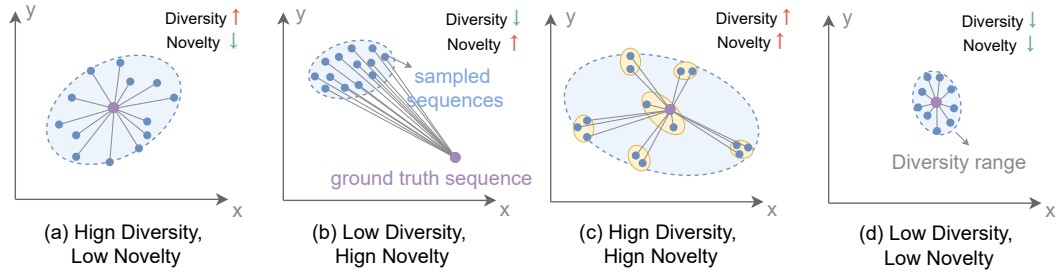

Figure 4: Conceptual illustration of the difference between diversity and novelty of the generated sequences. Each blue dot represents a generated sequence, and the purple dot represents the ground truth sequence. Assuming all generated sequences in this plane share similar structural consistency, The gray circular boundary indicates the diversity range among the generated samples, while the gray lines connecting each sample to the ground truth reflect their novelty.

**Implementation Details.** All experiments are conducted on NVIDIA-A100 GPUs with 80G memory. We choose ESM-3 [18] as our policy model because it is the first protein foundation model that directly supports inverse folding without task-specific fine-tuning. The Jumpy Denoising strategy also leverages it, which is capable of filling in arbitrary mask ratios. Building on this capability, we focus on unleashing the potential of PLMs through *test-time scaling*. To ensure fast structural feedback for reward computation, we use ESMFold [29] to predict the 3D structures of candidate sequences. For ProtInvTree, we set the maximum number of MCTS iterations $M$ to 50. The selection numbers $K_t$ at each step follow a cosine schedule. In the UCT algorithm, the weight $w$ balancing the exploration and exploitation is set to 0.01.

## 5.2 Benchmarking Fixed Backbone Protein Design

**Structural Consistency.** We benchmark the fixed backbone protein design task in CATH4.2 and CATH4.3 datasets, reporting the sc-TMscore and RMSD in Tables 1 and 2. ProtInvTree demonstrates superior performance over previous methods. We highlight the following: (1) Although iterative refinement models have significantly outperformed previous autoregressive and one-shot baselines, the proposed tree-based generation framework (ProtInvTree) further achieves substantial improvements, demonstrating the effectiveness of branching exploration over linear refinement. (2) ProtInvTree enhances inference based on the frozen ESM-3 model, requiring no additional trainable parameters, yet achieving the strongest performance. Compared to plain ESM-3 run for the same number of denoising steps (no tree search), ProtInvTree improves the sc-TMscore by 18.3% (short), 17.6% (single-chain), and 7.8% (all) in CATH 4.2. (3) We further evaluate RMSD as an independent structural metric, which consistently supports the effectiveness of ProtInvTree.

**Balance between Structural Consistency and Diversity & Novelty.** Figure 3 illustrates the Pareto frontier between structural consistency (measured by sc-TMscore) and two key sequence-level objectives: *diversity* (left) and *novelty* (right). We highlight our primary findings as follows: (1) ProtInvTree achieves **Pareto-optimal performance** in both the diversity–scTMscore and novelty–scTMscore

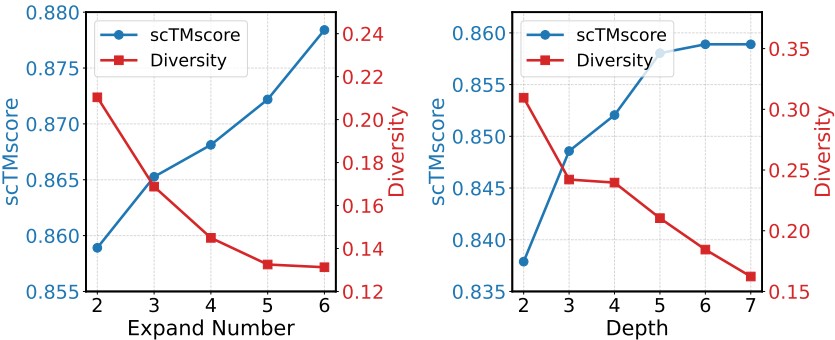

Figure 5: Test-time scaling laws analysis of our ProtInvTree under different expansion numbers (left) and search depths (right).

spaces, outperforming all baselines across the trade-off frontier. (2) **Compared to ESM-3**, we observe that ProtInvTree also achieves significantly higher diversity and novelty even at comparable levels of structural consistency. (3) Notably, when at comparable sc-TMScore levels, the baselines of AlphaDesign, PiFold, and KW-Design exhibit progressively *higher diversity and lower novelty*. This highlights a fundamental distinction between the two metrics: **diversity** measures variation within the set of generated sequences, whereas **novelty** reflects deviation from the native (ground-truth) sequence. As shown in Fig. 4, baseline methods optimized with a recovery loss tend to converge around local optima near the ground-truth sequence, as illustrated in case (a); by contrast, our method can escape this regime and explore multiple diverse and structurally consistent solutions, including those far from the ground-truth sequence, as shown in case (c).

### 5.3 De Novo Proteins Design

Evaluating models on the TS45 dataset allows us to gain a better understanding of the potential of AI models in designing de novo proteins and reveals that different models exhibit non-trivial differences in generalizability. We present the quantitative results in Table 3, which reveal the following: (1) **ProtInvTree** outperforms all baseline methods in terms of both sc-TMscore and RMSD, highlighting its superior ability to maintain structural consistency and achieve accurate geometric reconstruction. (2) We additionally compare ProtInvTree with ESM-3 [18] to assess the effectiveness of our overall framework beyond the pretrained language model itself. Despite sharing the same pretrained model, ProtInvTree achieves substantially better results, suggesting that test-time reward-guided planning plays a key role in unlocking the full potential of pretrained PLMs.

### 5.4 Analysis: Diving Deep into ProtInvTree

**Test-time Scaling Analysis.** To understand how test-time computation scales with performance, we investigate the effect of two key planning hyperparameters in our framework: *the number of candidate expansions* and *planning depth*, as shown in Fig. 5. We observe that, as the expansion number and planning depth increase, sc-TMscore gradually improves, although the average time cost also rises moderately. This indicates that scaling test-time computation can effectively enhance sequence quality through more deliberate search. However, as the number of planning depths further increases, the sc-TMscore tends to be saturated, as the search converges to high-confidence regions, the diversity of refinable sequences becomes limited, and further refinements yield diminishing structural gains. Moreover, the diversity in both settings decreases as the expansion number and planning depth increase, revealing the trade-off between structural consistency and sequence diversity.

**Computational Efficiency** The analysis of test-time computational efficiency is critical, particularly for methods that involve large pretrained models and structure prediction. We provide a detailed analysis of the compute–performance trade-off. Specifically, for each given backbone, we generate 10 designs and compute the average generation time for each design. We fix the depth and scale the expand number from 2 to 6, as shown in Table 4. We also scaled the number of sampling iterations for ESM-3 to assess its performance-efficiency trade-off. The resulting sc-TMScore and average inference time per design are summarized in Table 5. We observe that, although inference

time increases with a larger expand number or planning depth, the performance–efficiency trade-off remains superior to simply scaling the base ESM-3 sampler.

Table 4: The performance-efficiency trade-off of ProtInvTree (Our method)

| Expand Number | 2 | 3 | 4 | 5 | 6 |
|---|---|---|---|---|---|
| sc-TMcore | 0.859 | 0.865 | 0.868 | 0.872 | 0.878 |
| Inference Time / design (s) | 6.72 | 11.43 | 21.20 | 32.29 | 43.92 |

Table 5: The performance-efficiency trade-off of ESM3 (base model)

| Iteration Number | 5 | 10 | 50 | 80 | 100 |
|---|---|---|---|---|---|
| sc-TMScore | 0.816 | 0.822 | 0.825 | 0.824 | 0.826 |
| Inference Time / design (s) | 0.438 | 0.706 | 3.397 | 5.481 | 6.897 |

**Case Study.**  To facilitate understanding of the entire workflow of our proposed ProtInvTree, we visualize a reward-guided search tree in Figure 6. Each node represents a partially generated sequence with its predicted reward $r$, reflecting structural consistency. The tree showcases how ProtInvTree performs branching exploration guided by reward scores. Two high-reward sequences, S1 and S2, emerge from different trajectories, with high diversity and novelty, yet achieve high structural rewards ($r = 0.99$, $r = 0.98$). We further compare their predicted 3D structures with the ground truth structure in Figure 7. Its high sc-TMScore and low RMSD demonstrate **ProtInvTree**'s ability to generate diverse sequence candidates while maintaining structural consistency. This case illustrates how the reward-guided tree search enables efficient exploration of the solution space and selection of structurally faithful, non-trivial designs beyond native recovery.

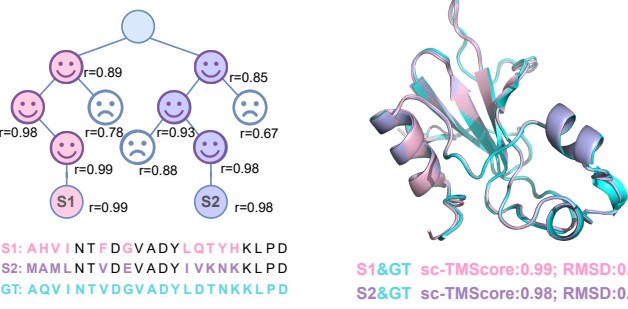

Figure 6: Reward-guided search tree visualization.

Figure 7: Structural alignment visualization.

## 6   Conclusion

We present ProtInvTree, a novel reward-guided tree-search framework for protein inverse folding that explicitly addresses the trade-off between structural consistency and sequence diversity. By reformulating sequence design as a step-wise, decision-making process, ProtInvTree enables the exploration of diverse design trajectories through self-evaluation, lookahead, and backtracking. ProtInvTree shows superior performance across multiple benchmarks, achieving state-of-the-art structural consistency while generating diverse and novel sequences beyond the native ground truth. Future work will focus on extending our framework to a broader range of protein-related tasks beyond fixed-backbone inverse folding. One limitation of the proposed ProtInvTree is the computational efficiency, particularly for methods that involve large pretrained models and structure prediction. Another potential limitation is its current lack of experimental validation in real-world biological settings. We will seek collaborations with experimental laboratories to test the viability and functional relevance of the designed sequences.

## Acknowledgments

This work is partially supported by the program of The Robotic AI-Scientist Platform of Chinese Academy of Sciences.

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

## A  Algorithms

The overall workflow of the ProtInvTree is provided in Algorithm 1.

---

**Algorithm 1** ProtInvTree: Reward-Guided Tree Search for Protein Inverse Folding

---

**Input:** Backbone structure $\mathbf{c}$, ground truth sequence $\mathbf{x}_{gt}$, initial sequence $\mathbf{x}_0$, folding model $f(\cdot)$, PLM policy $\pi_\theta$, reward function $R(\cdot, \cdot)$, max iterations $M$, tree depth $T$, expansion number per node $K$, reward threshold $\tau$

**Output:** A set of generated sequences $\mathcal{S} = \{\mathbf{x}_i^*\}_{i=1}^Z$, where $Z$ denotes the number of generated sequences.

1: Initialize root node $\mathbf{s}_0 = (\mathbf{c}, \mathbf{x}_0)$, search tree $\mathcal{T}$ with $\mathbf{s}_0$, and result set $\mathcal{S} \leftarrow \varnothing$
2: **for** $m = 1$ to $M$ **do**
3:    **Selection:** Traverse the tree from $\mathbf{s}_0$ using UCT to select a promising node $\mathbf{s}_t$ (Eq. 3)
4:    **for** $k = 1$ to $K$ **do**
5:       **Expansion:**
6:          Sample action $\mathbf{a}_t^{(k)} \sim \pi_\theta(\mathbf{a}_t \mid \mathbf{s}_t)$         $\triangleright$ Focus-and-Grounding strategies (Sec. 4.3)
7:          Apply $\mathbf{a}_t^{(k)}$ to obtain updated sequence $\mathbf{x}_{t+1}^{(k)}$
8:          Construct child state $\mathbf{s}_{t+1}^{(k)} = (\mathbf{c}, \mathbf{x}_{t+1}^{(k)})$
9:          Add $\mathbf{s}_{t+1}^{(k)}$ to tree $\mathcal{T}$ as child of $\mathbf{s}_t$
10:      **Evaluation:**
11:         Sample completed sequence: $\tilde{\mathbf{x}}_T^{(k)} \sim \mathcal{J}(\mathbf{x}_{t+1}^{(k)}, \mathbf{c})$         $\triangleright$ Jumpy denoising (Sec. 4.4)
12:         Compute reward: $r_{t+1}^{(k)} = \text{TMScore}(f(\tilde{\mathbf{x}}_T^{(k)}), f(\mathbf{x}_{gt}))$
13:         Set node value: $V(\mathbf{s}_{t+1}^{(k)}) = r_{t+1}^{(k)}$
14:      **Backpropagation:**
15:         Update visit count $N(\mathbf{s}_j)$ and value $V(\mathbf{s}_j)$ (Eq. 5)
16:         Backpropagate $r_{t+1}^{(k)}$ to update all ancestors of $\mathbf{s}_{t+1}^{(k)}$ (Eq. 6)
17:      **if** $t + 1 = T$ **and** $r_{t+1}^{(k)} \geq \tau$ **then**
18:         Add $\mathbf{x}_{t+1}^{(k)}$ to result set $\mathcal{S}$
19:      **end if**
20:    **end for**
21: **end for**
22: **Return:** Sequence set $\mathcal{S} = \{\mathbf{x}_i^*\}_{i=1}^Z$ containing $Z$ high-quality candidates

---

## B  Evaluation Metrics

In the main paper, we report evaluation results using four metrics: sc-TMscore, RMSD, novelty, and diversity. The descriptions of these metrics are detailed as follows.

**sc-TMScore.** The structural similarity is the ultimate standard for measuring the quality of the designed sequence. However, the structures of designed protein sequences needed to be predicted by other algorithms, such as AlphaFold [1], RoseTTAFold [3], OmegaFold [52] and ESMFold [29]. The protein folding algorithm itself has a certain inductive bias and will cause some prediction errors, which will affect the evaluation. To overcome the inductive bias, we adapt the self-consistent TM-score (sc-TMscore) metric:

$$\text{sc-TMScore} = \text{TMScore}(f(\tilde{\mathbf{x}}), f(\mathbf{x})), \tag{12}$$

where $f$ is the protein folding algorithm and $\text{TMScore}(\cdot, \cdot)$ is a widely used metric [60] for measuring protein structure similarity. Since the structures of the designed sequence and reference sequence are predicted by the same protein folding algorithm, the model's inductive bias is expected to be canceled out when calculating the TM-score. This approach results in a more robust metric, called the sc-TMScore, that is less affected by the inductive bias of the protein folding algorithm.

**RMSD.** The standard dissimilarity measure for protein structures is the root mean square deviation (RMSD) of representative atom positions such as $\alpha$-carbons. RMSD is calculated as the square root

of the average squared distance between corresponding atoms in two 3D structures:

$$\text{RMSD}(\mathbf{v}, \mathbf{w}) = \sqrt{\frac{1}{n} \sum_{i=1}^{n} \|v_i - w_i\|^2}, \tag{13}$$

where $\mathbf{v} = f(\tilde{\mathbf{x}})$ and $\mathbf{w} = f(\mathbf{x})$ are the predicted 3D structures of the designed sequence $\tilde{\mathbf{x}}$ and the reference sequence $\mathbf{x}$, respectively, obtained using a structure prediction algorithm $f$. Here, $v_i$ and $w_i$ denote the 3D coordinates of the $i$-th atom in each structure, and $n$ is the total number of atoms considered (typically backbone or $\alpha$-carbon atoms). RMSD provides a fine-grained comparison of atomic positions after optimal rigid-body alignment of the two structures. However, it is sensitive to local deviations, such as flexible loops or inaccurate predictions in side-chain packing, and may not fully reflect the overall fold similarity. As a result, RMSD is typically used in conjunction with other metrics such as TM-score to provide a more comprehensive assessment of structural quality.

**Novelty.** We define novelty as the complement of sequence recovery, reflecting the extent to which the generated sequences deviate from the native ground truth:

$$\text{Novelty} = 1 - \text{Recovery}$$

where recovery is the fraction of amino acids in the predicted sequence that exactly match the ground-truth sequence at each position, defined as:

$$\text{Recovery} = \frac{1}{n} \sum_{i=1}^{n} \mathbb{1}(\tilde{x}_i = x_i)$$

**Diversity.** The average fraction of amino acids that differ between pairs of sequences:

$$\text{Diversity}(\{\tilde{\mathbf{x}}^1, \ldots, \tilde{\mathbf{x}}^M\}) = \frac{2}{NM(M-1)} \sum_{j=1}^{M} \sum_{k=1}^{j-1} \sum_{i=1}^{N} \mathbb{1}[\tilde{\mathbf{x}}^j[i] \neq \tilde{\mathbf{x}}^k[i]]$$

$$= \frac{2}{M(M-1)} \sum_{j=1}^{M} \sum_{k=1}^{j-1} d_H(\tilde{\mathbf{x}}^j, \tilde{\mathbf{x}}^k).$$

where $d_H$ is the Hamming distance. We note that sequence diversity alone is not a sufficient measure of an IF method's quality, as it can be increased arbitrarily at the expense of sample quality (e.g. as measured by structural consistency).

## C   Analysis of the Planning Components

We investigate how explicit planning components—*intermediate feedback*, *uncertainty tracking*, and *backtracking*—affect test-time protein design quality. We compare our MCTS-based planner (ProtInvTree) against two commonly used test-time strategies: Best-of-$N$ and Beam Search sampling.

- **ESM-3**: base model without any planning or feedback;
- **Best-of-$N$**: generates $N$ independent candidates and returns the one with the highest reward;
- **Beam Search**: incrementally expands partial solutions while keeping only the top-$B$ (beam width) candidates, iterating step-by-step with intermediate feedback until a complete sequence/design is found;
- **ProtInvTree (ours)**: Monte Carlo Tree Search with intermediate feedback, uncertainty tracking, and backtracking.

Table 6 summarizes the results and the presence of each planning component. We find that (1) **Intermediate feedback** (Beam Search) improves convergence over purely independent sampling (Best-of-$N$), indicating that partial evaluations guide more promising local expansions. (2) **Uncertainty tracking** further boosts reliability by prioritizing candidates with stable scores, reducing variance in final structures. (3) **Backtracking** enables recovery from suboptimal branches and better global exploration, yielding the best overall accuracy.

Table 6: Comparison of planning strategies under equal computational budgets. Checks (✓) indicate the presence of the corresponding planning component.

| Models | sc-TMscore (↑) | RMSD (↓) | Intermediate Feedback | Uncertainty Tracking | Backtracking |
|---|---|---|---|---|---|
| ESM-3 | 0.816 | 2.135 | × | × | × |
| Best-of-$N$ | 0.839 | 1.974 | × | × | × |
| Beam Search | 0.853 | 1.724 | ✓ | × | × |
| ProtInvTree | **0.881** | **1.513** | ✓ | ✓ | ✓ |

## D    Selection Strategies Comparison

To analyze the impact of different position selection strategies in the *Focus-and-Grounding* action mechanism, we evaluate several variants for computing the position distribution $p_\phi(i \mid \mathbf{s}_t)$, which determines the set of positions $\{i_1, \ldots, i_{K_t}\}$ to be modified at each denoising step.

Specifically, we compare the following approaches:

- **Random sampling:** Positions are selected uniformly at random from the sequence.
- **Autoregressive sampling:** Positions are visited sequentially from left to right in an autoregressive manner.
- **Entropy-based selection:** Positions with the lowest predictive entropy, representing the model's most confident predictions, are prioritized for update.

We integrate each strategy into the *Focus* module $\mathcal{F}(\mathbf{s}_t)$, keeping the *Grounding* step unchanged. Table 7 summarizes the quantitative results, showing that all three strategies achieve competitive performance, with **random sampling** performing surprisingly well despite of its simplicity. This may be because exploring a broader space in the early stages helps avoid premature convergence and encourages greater sequence diversity, which ultimately benefits overall generation quality.

Table 7: Comparison of different sampling strategies on structure consistency (scTM-score).

| Sampling Strategy | Random | AR | Entropy |
|---|---|---|---|
| sc-TMscore (↑) | **0.881** | 0.877 | 0.870 |

## E    Structure Complexity Effects

We also provide additional breakdowns across fold classes and protein lengths, revealing meaningful differences in performance and novelty trends. Table 8 summarizes the breakdown by fold class and length-based results are given in Table 1.

Table 8: Performance trends across the Fold class.

| Fold Class | sc-TMscore (↑) | RMSD (↓) | Novelty (↑) |
|---|---|---|---|
| Mainly Alpha | 0.855 | 1.548 | 0.616 |
| Mainly Beta | 0.830 | 1.710 | 0.503 |
| Alpha-Beta | 0.908 | 1.488 | 0.448 |
| Few Secondary | 0.764 | 1.349 | 0.553 |

## F    Broader Impacts

Inverse protein folding models, positioned at the intersection of bioinformatics and computational biology, offer significant potential for advancing both basic research and real-world applications. By enabling the design of protein sequences that reliably fold into desired three-dimensional structures, these models can drive progress across diverse domains. Broader impacts include facilitating structure-based drug discovery, enabling the rational design of enzymes with novel functionalities, and advancing synthetic biology through the creation of custom proteins with tailored properties.

