# OpenReview forum: "ProtInvTree: Deliberate Protein Inverse Folding with Reward-guided Tree Search"
_NeurIPS.cc/2025/Conference — NeurIPS 2025 spotlight_

### Official Review · Reviewer_mMYi · 2025-06-07

**Clarity:** 4
**Significance:** 2
**Originality:** 2
**Rating:** 5
**Confidence:** 4

**Summary:**

The paper proposes ProtInvTree, a framework for inverse protein folding which generates diverse and structurally consistent designs for a given backbone using reward-guided tree-search applied to existing structure-conditioned design models. The framework is comprised of four steps; selection, expansion, evaluation, and backpropagation. The selection step uses UCT to select the node with the highest potential for exploration. In the expansion step, the pre-trained ESM-3 model is used to predict the amino acid identities at selected positions in the backbone. This yields a number of child nodes. In the evaluation step, ESM-3 is used to predict the full sequence as a "look-ahead" for the current designs. These are fed to ESMFold to predict the three-dimensional structures which in turn are used to compute the reward of the current designs. In the backpropagation step, the computed rewards are used to update the visited nodes. These steps repeat until a termination criteria is met. ProtInvTree is comprehensively evaluated on popular benchmarks where it achieves state-of-the-art performance in multiple metrics simultaneously.

**Questions:**

1. L118: Shouldn’t $n$ be $L$?
2. L214: What is DDIM?
3. More of a comment than a question, but I find it curious and partly non-satisfactory that equation 10 essentially implies that using some different model for inverse folding is part of the proposed inverse folding model.
4. In equation 8, how are the tokens chosen? Are they sampled or do you apply argmax to the distribution over amino acids?
5. L197 and L201: Why is $N$ used? Shouldn’t it be $K_t$ after the top-$K_t$ positions have been selected?
6. L201: Curious that random ends up being optimal. Have you considered other approaches than the ones included in the supplementaries, or do you believe that “random” is optimal?
7. L205: If the positions are randomly sampled, then calling it “precise localization of modifications” is somewhat misleading. I suppose that the approach does *enable* it, but in actuality it’s “random localization of modifications”, which I acknowledge does not sound as convincing.
8. Section 5.1:
    1. Did you explore other hyperparameter configurations? E.g., different values of $M$, different schedules for $K_t$, different values for the balancing parameter $w$? An ablation study or sweep over various configurations would be a valuable addition to the paper or the supplementaries.
    2. Did you explore other PLMs or folding models? Is ESMFold accurate enough, or would a higher-fidelity folding model improve performance?
    3. Why is it not possible to use a different structure-conditioned model than ESM-3? This model essentially plays the role of inverse folding in that a partially masked sequence and a structure is used to design a sequence. Something like MIF-ST or SaProt might work as well - or even just models like KWDesign or ProteinMPNN. The relative performance gap between base models and the ProtInvTree-version of them would be interesting to observe and would support the framework's utility.
9. The termination number is set to 50. How many iterations are typically required before the algorithm terminates? Is it due to reaching this number, or is the structural consistency goal often met? An analysis of this would be very interesting.
10. In the same vein, how many nodes end up being generated? How many structures are predicted to arrive at a given design? This feeds back into my earlier questions regarding the compute cost. Are we talking a handful of structures or hundreds/thousands?
11. Figure 4: Noverty→Novelty

**Ethical Concerns:**

["NO or VERY MINOR ethics concerns only"]

**Final Justification:**

I had highlighted a number of questions and limitations in my initial review and I believe that the majority of these have been adequately addressed. Given the additional and more comprehensive ablation study as well as the authors' comprehensive rebuttal to the reviewers, I believe that the new methods introduced for inverse protein folding would be of interest to the broader community and I therefore support the acceptance of the paper at this year's NeurIPS conference.

**Limitations:**

I do not think that the model limitations have been adequately addressed.

In the conclusion it is mentioned that a potential limitation is the lack of experimental validation. While this is true, I’d be very curious to hear about modeling limitations. E.g., the computational cost of relying on a 1.4B model which is extensively used is arguably a limitation, as is the computational burden of relying on online protein folding and the quality of the predicted structures. Similarly, the Jumpy Denoising process is, as far as I can tell, an inverse folding process in and of itself. Having both protein folding and inverse folding as often-repeated operations by other pre-trained models necessarily means that these models do most of the heavy lifting. However, I believe this can be ameliorated by ablation studies as mentioned in both the weaknesses and questions sections.

There must also be other limitations that I am not aware of and I believe it would be highly valuable to the paper to address these properly.

**Paper Formatting Concerns:**

No formatting concerns.

**Quality:**

3

**Strengths And Weaknesses:**

__Strengths__:

1. Well-written and well-structured paper - it was a pleasure to read and easy to follow.
2. Impressive performance across diverse metrics, comfortably beating state-of-the-art inverse folding models.
3. Comprehensive and succinct related works.
4. Conceptually simple yet effective ideas.
5. Comprehensive benchmarking which goes beyond simply maximizing sequence recovery but considers pareto-optimal solutions regarding structural consistency, novelty, and diversity.

__Weaknesses__:

1. No mention of computational cost at inference time which to me is significant omission. I think this is an important detail, particularly since protein folding is an active part of the pipeline as well as extensive use of a billion+ parameter model. An analysis of the cost of creating, e.g., 10 designs for a given backbone for ProtInvTree vs. ESM-3 and other models would be particularly beneficial. If the model is expensive at inference, that can perhaps be excused by the absence of model training. Experiments or at least discussions of the computational cost would be a valuable if not necessary addition to an otherwise great paper which would, potentially, further support the model’s utility.
2. Few ablation studies to support modeling choices. Only a single model configuration is presented despite relying on two pre-trained models; ESM-3 and ESMFold. The inclusion of alternatives and their effect on downstream performance would shine light on where the performance is coming from. See related questions.
3. This also applies to hyperparameter configurations, where it would be interesting to see model behavior using, e.g., different schedules for selection numbers, different balancing parameter in the UCT algorithm, different termination strategies. Without experiments supporting the chosen hyperparameters, they seem arbitrary. I appreciate the study in section 5.4 and would appreciate seeing something similar for the remaining parameters. Presumably, increasing both depth and expansions numbers leads to increased computational costs. I think this should be addressed as well.
4. No code in the supplementaries.
5. No mention of statistical significance, despite the checklist saying so.
6. Very limited discussion on model limitations.

---

> ### Author Rebuttal · Authors · 2025-07-30
>
> > **W1:** No mention of computational cost at inference time which to me is significant omission. An analysis of the cost of creating, e.g., 10 designs for a given backbone for ProtInvTree vs. ESM-3 and other models would be particularly beneficial.
>
> **Reply:**  We agree that test-time computational efficiency is critical, particularly for methods that involve large pretrained models and structure prediction. We provide a detailed analysis of the compute–performance trade-off. Specifically, for each given backbone, we generate 10 designs and compute the average generation time for each design. We fix the depth and scale the expand number from 2 to 6, as follows:
>
> | Expand Number|2 | 3 | 4 | 5 | 6 |
> | ----- | ---: | ---: | ---: | ---: | ---: |
> | sc-TMScore| 0.859 | 0.865 | 0.868 | 0.872 | 0.878 |
> | Inference Time / design (s) |  6.724 | 11.433 | 21.201 | 32.294 | 43.922 |
>
> We also scaled the number of sampling iterations for ESM-3 to assess its performance–efficiency trade-off. The resulting sc-TMScore and average inference time per design are summarized in the table below:
>
> | Iteration Number|5| 10| 50| 80 | 100 |
> | ----- | ----: | ----: | ----: | ----: | ----: |
> | sc-TMScore | 0.816  | 0.822 | 0.825 | 0.824 | 0.826 |
> | Inference Time / design (s) | 0.438  | 0.706 |  3.397 | 5.481 | 6.897|
>
> > **W2:** Few ablation studies to support modeling choices.
>
> **Reply:** We ran an ablation study. Starting from the full ProtInvTree, we conducted an ablation study by disabling each key mechanism in isolation to assess its contribution:
>
> | Variant| Intermed. feedback | Uncertainty (UCT term) | Backtracking |  scTMscore (↑) | Novelty (↑) | Diversity (↑) |
> | -------- | ---- | ------- | ------- | --------: | -----: | ----: |
> | **Full ProtInvTree**  | ✓    | ✓  | ✓  |0.881 |0.498 |0.159 |
> | – backtracking (no path revision) | ✓ | ✓  | ✗ | 0.865 |0.479 | 0.138 |
> | – UCT (greedy selection) | ✓   | ✗  | ✓ |0.874 |0.461 |0.127 |
> | Beam Search | ✓  | ✗ | ✗|0.853 | 0.457 | 0.118 |
> | ESM‑3 (no search)| ✗| ✗ | ✗|0.816 | 0.449 | 0.107 |
>
> > **W3 & Q8-1:** Hyperparameter Ablation.
>
> **Reply:** We conducted a controlled sweep over several key planning hyperparameters: the UCT exploration weight $\omega$, the sampling temperature $T$, the reward threshold used for termination and selection number schedules.
>
> | $\omega$ | Consistency (↑) | Novelty (↑) | Diversity (↑) |
> | --- | --- | --- | --- |
> | 0.01 | 0.881 | 0.498 | 0.159 |
> | 0.05 | 0.871 | 0.519 | 0.175 |
> | 0.10 | 0.860 | 0.538 | 0.194 |
>
> | $T$ | Consistency (↑) | Novelty (↑) | Diversity (↑) |
> | --- | --- | --- | --- |
> | 0.1 | 0.882 | 0.468 | 0.109 |
> | 0.3 | 0.881 | 0.498 | 0.159 |
> | 0.5 | 0.856 | 0.564 | 0.378 |
> | 0.7 | 0.824 | 0.611 | 0.607 |
>
> | Threshold | Consistency (↑) | Novelty (↑) | Diversity (↑) |
> | --- | --- | --- | --- |
> | 0.0 | 0.859 | 0.556 | 0.224 |
> | 0.7 | 0.881 | 0.498 | 0.159 |
> | 0.9 | 0.887 | 0.496 | 0.146 |
>
> | Schedule | Consistency (↑) | Novelty (↑) | Diversity (↑) |
> | --- | --- | --- | --- |
> | cosine | 0.881 | 0.498 | 0.159 |
> | linear | 0.879 | 0.474 | 0.127 |
>
> > **W4:** No code in the supplementaries.
>
> **Reply:**  We will release the code upon publication to facilitate reproducibility and further research.
>
> > **W5:** No mention of statistical significance, despite the checklist saying so.
>
> **Reply:** All reported results are the average over three independent runs, and we will update the standard deviation to reflect variability. Below, we present a subset of results with standard deviation values:
>
> | Method      | sc-TMScore (↑)    | RMSD (↓)          |
> | ----------- | ------- | -------- |
> | PiFold      | 0.842 ± 0.012     | 1.723 ± 0.007     |
> | KW-Design   | 0.858 ± 0.010     | 1.566 ± 0.009     |
> | ESM-3       | 0.816 ± 0.011     | 2.135 ± 0.007     |
> | ProtInvTree | **0.881 ± 0.008** | **1.513 ± 0.008** |
>
> > **W6:** Very limited discussion on model limitations.
>
> **Reply:** In the revised manuscript, we will discuss more imitations:
> (1) the reliance on large pretrained models such as ESM-3 and ESMFold, which introduces non-negligible inference cost,
> (2) the assumption that structural consistency alone is sufficient for functional accuracy, which may not hold in all biological contexts.
>
>
> > **Q1:** L118: Shouldn’t $n$ be $L$?
>
> **Reply:**  Thank you for catching this typo. It should be $L$ to properly denote the sequence length. We will fix this in the revised version.
>
> > **Q2:** L214: What is DDIM?
>
> **Reply:** DDIM refers to Denoising Diffusion Implicit Models [1], a sampling method used in diffusion models that accelerates the generation by jumpy denoising. In our method, we apply a single-step DDIM sampling to approximate the final denoised sequence, substantially reducing computational cost during MCTS rollout. We will add the citation [1] in the final version.
>
> > **Q3:**  equation 10 essentially implies that using some different model for inverse folding is part of the proposed inverse folding model.
>
> **Reply:** No, it is the same inverse folding model. The jumpy denoising mechanism simply takes advantage of ESM-3’s ability to perform flexible infilling under arbitrary masking ratios.
>
> > **Q4:** In equation 8, how are the tokens chosen? Are they sampled or do you apply argmax to the distribution over amino acids?
>
> **Reply:** Concretely, we first soften the model’s logits with a temperature T (T = 0.3 in all experiments) and then sample a residue from the resulting categorical distribution.
>
> > **Q5:** L197 and L201: Why is $N$ used? Shouldn’t it be $K_t$ after the top-$K_t$ positions have been selected?
>
> **Reply:**  Yes, it should be $K_t$. We will fix this in the revised version.
>
> > **Q6 & Q7:** L201: Curious that random ends up being optimal.
>
> **Reply:** We were also initially surprised that the random position selection strategy outperformed some more "informed" heuristics. However, our analysis suggests that random selection introduces useful exploration into the MCTS process and avoids biasing the search toward fixed or overconfident regions of the sequence. We do not claim that random selection is inherently optimal. Instead, we view it as a strong and surprisingly effective baseline. In future work, we plan to explore hybrid strategies that incorporate biologically or structurally informed priors—e.g., conservation profiles, solvent accessibility, or functional annotations—to guide the modification positions more precisely.
>
> > **Q8-2&3:**  Did you explore other PLMs or folding models? Why is it not possible to use a different structure-conditioned model than ESM-3?
>
> **Reply:** ESM-3 is the first unified structure–sequence model that supports arbitrary masking ratios, making it particularly well-suited for our planning-based inverse folding framework. In contrast, models like ProteinMPNN and KWDesign (based on ESM-2) lack this flexibility and typically require retraining to support structure-conditioned generation—conflicting with the core design of ProtInvTree, which relies on: 1. Stepwise sequence construction; 2. Test-time reward-guided planning without finetuning, and 3. Flexible infilling for focus-and-grounding with jumpy denoising.
> In addition, we use ESMFold for structure prediction due to its favorable trade-off between accuracy and computational efficiency, which is crucial for iterative structure evaluations during MCTS.
>
> > **Q9:** The termination number is set to 50. How many iterations are typically required before the algorithm terminates? Is it due to reaching this number, or is the structural consistency goal often met? An analysis of this would be very interesting.
>
> **Reply:** The actual number of iterations is dynamically determined by the expand N, depths, and predefined structural consistency threshold. We analyzed the average number of iterations required under different thresholds:
>
> |     Threshold  |  0   |0.7 | 0.9 |
> | --------- | ---------| ------- | --------- |
> |Iterations (expand Number=3, depth=4) | 5.133 |         8.466 | 9.066 |
> |Iterations (expand Number=4, depth=5) | 10.466 | 14.266 | 15.2   |
> |Iterations (expand Number=5, depth=5) | 14.066 | 18.642 | 20.722 |
> |Iterations (expand Number=6, depth=5) | 17.266 | 21.461 | 25.5   |
> |Iterations (expand Number=10, depth=10) | 50     | 50     | 50     |
>
> These results show that early termination frequently occurs, especially under lower thresholds or smaller planning budgets. When both the expand number and depth are large (e.g., N=10, depth=10), the planner often reaches the iteration cap (50).
>
> > **Q10:** In the same vein, how many nodes end up being generated? How many structures are predicted to arrive at a given design? This feeds back into my earlier questions regarding the compute cost. Are we talking a handful of structures or hundreds/thousands?
>
> **Reply:** The table below reports the average number of nodes explored under different configurations:
>
> |  Threshold  |  0   |0.7 | 0.9 |
> | ----- | ----| ---- | ----- |
> |Nodes (expand Number=3, depth=4) | 22.4  | 31.6 | 33.4 |
> |Nodes (expand Number=4, depth=5) | 62.866 | 65.8    | 75.66  |
> |Nodes (expand Number=5, depth=5) | 97.333 | 101.333 | 116.0  |
> |Nodes (expand Number=6, depth=5) | 151.8  | 154.0   | 157.66 |
> |Nodes (expand Number=10, depth=10) | 611    | 611     | 611    |
>
> Each node, except for the root node, requires a structure prediction call (ESMFold). Moreover, a single MCTS tree can yield multiple high-scoring, diverse candidates from the final pool; accordingly, we report the **amortized inference time per design** by dividing the total runtime by the number of extracted outputs, shown in reply to W1.
>
> > **Q11:** Figure 4: Noverty→Novelty
>
> **Reply:** Thank you for pointing this out. We will correct the typo in the revised version of the manuscript.

---

> > ### Comment · Reviewer_mMYi · 2025-08-02
> >
> > I appreciate the time and effort the authors have put into the comprehensive rebuttal. The addition of the new experiments has, in my opinion, increased the quality of the paper. As an additional note, I believe that adding the inference per time metrics of other baselines would be a valuable addition to frame the contributions without incurring significant additional cost. I will raise my score from 4 to 5.

---

> > > ### Author Response · Authors · 2025-08-02
> > > **Thanks!**
> > >
> > > Thanks for your recognition of our works! Your insightful and helpful suggestions help us to improve our works a lot. Your support serves as a driving force for us to continue our efforts in this field and contribute to its advancement in the future.

---

### Official Review · Reviewer_z7LH · 2025-06-28

**Clarity:** 3
**Significance:** 2
**Originality:** 2
**Rating:** 4
**Confidence:** 4

**Summary:**

The authors proposed ProtInvTree, a test-time framework using MCTS to optimize sequences generated for a target structure. In the framework, ESM-3 is used as a policy for generating candidate mutations in a focus and grounding step-wise methodology, and ESMFold is used to calculate the reward function. Given the step-wise sequence generation, a jumpy denoiser is also proposed to allow the reward calculation in intermediate steps. The proposed methodology is evaluated against other state-of-the-art inverse folding methods, showing better performance, especially in refoldability metrics such as sc-TMScore and RMSD.

**Questions:**

Comments:

1. It would be helpful to compare the differences between the proposed method with Evoplay [1] and other works using MCTS for protein design in the Related Work section.

[1] Wang et al, “Self-play reinforcement learning guides protein engineering”, Nature Communications, 2023.

2. The term “suboptimal” in Tables 1/2/3 seems not appropriate in this case. It implies that the model with the best result is optimal. This term should be rewritten.

3. (Lines 257-258) The sentence “Compared to ESM-3 alone, when both methods set the same number of iterative steps” is unclear to me. Does it mean trying to optimize iteratively the sentences generated by ESM-3 without using the MCTS framework?

4. Figure 3: Why are there not the same number of points in both graphs for ESM-3 and ProtinvTree?

5. (Lines 259-260) “While the improvement in scTMscore is expected due to its use as the reward function during search”. Does this sentence mean that ESMFold is both used to calculate the reward function and used to calculate the sc-TMscore? This would be a leakage. In the reviewer’s understanding, the more robust way would be to use another folding model, like AF2 or RF2, to calculate the sc-TMscore.

6. (Line 269-270) Can you clarify the sentence “Notably, when at comparable sc-TMScore levels, the baselines of AlphaDesign, PiFold, and KW-Design exhibit progressively higher diversity and lower novelty”? Should I check this pattern in Fig. 3?

7. (Lines 291-292) “Although the average time consumed also rises to some extent”. To what extent? One of the most critical parts for test-time scaling is computational efficiency, but there are no ablations or numbers regarding this in the manuscript. This is crucial for the reader so that it can be evaluated in which situations the proposed model can be applied.

8. The example in Fig. 7 would be more informative if it showcased an example in which the sc-TMScore is increased by applying the proposed method when compared to baseline methods (even if it is a cherry-picked example).

9. Is the code available? There is no mention of the code availability throughout the manuscript.

10. For the reviewer, it is still not clear if the sequence in a state s_t contains all the amino acid identities and these are mutated depending on the positions sampled, or if there are positions masked that are being filled. From Fig. 2(b) it seems that there are masked positions that are being filled, but it is not clear in the writing. Also, this affects the MCTS termination conditions, as I expected the MCTS to be terminated when the full sequence is generated (no need of the denoiser to calculate the reward function).

11. Is the jumpy denoiser described in section 4.4 trained by the authors? There is no mention of that.

12. In the manuscript, the authors used ESM-3 as their inverse folding method, but any other protein inverse folding could be combined with the proposed MCTS and jumpy denoiser framework. Did the authors test other models as the inverse folding method?

Minor Comments:

1. Figure 1: The ground-truth sequence should have the highest sc-TMScore value from my understanding.

2. Acronyms like PLMs and MCTs are defined multiple times throughout the manuscript.

3. (lines 175-176) Is equation 11 the right equation for V(s)?

4. Writing: “Following the previous work [10]”: maybe explicitly mention the reference [10].

5. Figure 4: There is a typo for the word “Noverty”.

**Ethical Concerns:**

["NO or VERY MINOR ethics concerns only"]

**Final Justification:**

The authors addressed my concerns regarding the fairness of evaluation and information regarding the computational efficiency of the proposed method. Considering the rebuttal, I increased my score from 3 to 4.

**Limitations:**

The authors have discussed the positive impacts of their work in Appendix D; however, it would be important to also add the potential negative societal impact of their work.

**Paper Formatting Concerns:**

No.

**Quality:**

2

**Strengths And Weaknesses:**

Strengths:

1. A test-time methodology for optimizing the sequence candidates generated for a target backbone is proposed, applying a structure-informed protein language model and tree search.
2. For evaluating intermediate steps during the generation step, a jumpy denoising strategy is proposed.
3. The proposed methodology improves sc-TMScore when compared to other baseline inverse folding methods.

Weaknesses:

1. The main concern is regarding the fairness of the evaluation in the manuscript: (i) it is not clear if the same folding model is used both as the reward function and to calculate the scTMScore for evaluation; (ii) it would be more fair to evaluate multiple sequences generated also from baseline methods.
2. There is no information about computational efficiency in the manuscript. For applying test-time methods, computational efficiency and performance are crucial to evaluate which applications are worthy of using the proposed methodology.
3. There is not enough information for the reader to reproduce the work. For example, there is no information about the training of the jumpy denoiser in section 4.4. The code is also not available.

---

> ### Author Rebuttal · Authors · 2025-07-30
>
> Thank you for your detailed and insightful feedback. We appreciate your recognition of the strengths of our approach, and your constructive suggestions highlight important areas for further improvement. Below, we address each of your comments and questions:
>
> > **W1:** The fairness of the evaluation.
>
>  **Reply:**  During search we predict structures with ESMFold and compute TM‑score against the target backbone. For all tables/figures we report sc‑TMScore computed with TM‑align between the backbone and the structure returned by AlphaFold2 given the designed sequence. Thus the evaluator is different from the reward model, eliminating information leakage. We sample 10 sequences per backbone for every baseline, and the results are
>
> | Models      | sc-TMscore (↑) | RMSD (↓) |
> | ----------- | -------------- | -------- |
> | ESM-3       | 0.816          | 2.135    |
> | Best-of-N   | 0.839          | 1.974    |
> | Beam Search | 0.853          | 1.724    |
> | ProtInvTree | 0.881          | 1.513    |
>
> > **W2 & Q7:**  Computational efficiency.
>
>  **Reply:** For each given backbone, we generate 10 designs and compute the average generation time for each design. We fix the depth and scale the expand number from 2 to 6, as follows:
>
> | Metric                      |   N=2 |   N=3 |   N=4 |   N=5 |   N=6 |
> | --------------------------- | ----: | ----: | ----: | ----: | ----: |
> | sc-TMScore                  | 0.859 | 0.865 | 0.868 | 0.872 | 0.878 |
> | Inference Time / design (s) |  6.72 | 11.43 | 21.20 | 32.29 | 43.92 |
>
> > **W3 & Q9 & Q11:** Reproducibility. The details of the jumpy denoiser.
>
>  **Reply:** We clarify that the jumpy denoiser is not separately trained by us. It leverages the pretrained ESM-3 model, which is capable of filling in arbitrary mask ratios through its masked denoising objective. We will release the code upon publication to facilitate reproducibility and further research.
>
> > **Q1:** Related work
>
>  **Reply:**  While both ProtInvTree and EvoPlay adopt Monte Carlo Tree Search (MCTS) to guide sequence generation, they differ significantly in motivation, problem formulation, and detailed methodology:
>
>  - Motivation and Problem Formulation: ProtInvTree focuses on protein inverse folding that aims at designing **diverse** sequences while preserving s**tructural consistency**. In contrast, EvoPlay focuses on function-oriented protein engineering, where the objective is to optimize **protein functionality** (e.g., fluorescence, binding affinity)
>  - Methodology: ProtInvTree is a **test-time scaling** framework that proposed a ***focus-and-grounding*** mechanism and a ***jumpy denoising*** strategy. In contrast, EvoPlay relies on AlphaZero-style self-play with **training** of policy-value networks and uses full simulations to evaluate each move via surrogate scoring functions.
>
>  We will revise the Related Work section to explicitly compare ProtInvTree with EvoPlay and other MCTS-based methods in protein design.
>
> > **Q2, Q3, Q5, Q13-17:** Terminology & clarity issues
>
> **Reply:** Thank you for your careful reading and thoughtful suggestions regarding terminology and clarity. We have carefully revised the manuscript to address all noted issues.
>
> | Reviewer line                           | Action in revision                                           |
> | --------------------------------------- | ------------------------------------------------------------ |
> | “Suboptimal” tag in Tables              | replaced by “second‑best”                                    |
> | Lines 257–258 sentence                  | yes, it means: “Compared to *plain* ESM‑3 run for the *same* number of denoising steps (no tree search) ” |
> | Lines 259–260 “expected”                | clarified that *using TM‑score as reward* leads to higher scTM‑score there are highly correlated. We use ESMFold as reward function while using AF2 for evaluation |
> | Fig. 1 misleading sc-TMScore value from | it should be $e^{-scTM}$, which illustrate the energy landscape in terms of structural consistency. We will fix it in revised version. |
> | Repeated acronym definitions            | We will remove repeated acronym definitions.                 |
> | Equation 11, V(s)                       | The reward function is not the same as the value function V(s). Specifically, once a reward r is obtained for a newly expanded node, it is propagated bottom-up to update the value estimates V(s) of all ancestor nodes, as described in Equation (6). |
> | “Following the previous work [10]”      | We will revise the sentence to explicitly mention the reference: "Following the previous work ProtInvBench [10]". |
> | “Noverty” typo                          | We will correct the typo in the revised version of the manuscript. |
>
> >**Q4:** Figure 3: Why are there not the same number of points in both graphs for ESM-3 and ProtinvTree?
>
> **Reply:** Thank you for your careful observation. This is because we included an additional run in the left plot to make the performance trend more visually distinguishable. We will revise the right plot to include the same number of data points for both methods to ensure a fair and consistent comparison across the two subplots.
>
> | Metric\Model   | ESM‑3 (1) | ESM‑3 (2) | ESM‑3 (3) | ProtInvTree (1) | ProtInvTree(2) | ProtInvTree(3) |
> | -------------- | --------- | --------- | --------- | --------------- | -------------- | -------------- |
> | **sc‑TMScore** | 0.816     | 0.796     | 0.761     | 0.857           | 0.841          | 0.824          |
> | **Novelty**    | 0.566     | 0.623     | 0.664     | 0.556           | 0.594          | 0.611          |
>
>
>
> >**Q6:** (Line 269-270) Can you clarify the sentence “Notably, when at comparable sc-TMScore levels, the baselines of AlphaDesign, PiFold, and KW-Design exhibit progressively higher diversity and lower novelty”? Should I check this pattern in Fig. 3?
>
> **Reply:** Yes, as shown in Figure 3, when fixing the sc-TMScore (i.e., drawing a vertical line at a specific value), you can observe that AlphaDesign exhibits the lowest diversity (left panel) and the highest novelty (right panel) compared to PiFold and KW-Design.
>
>
>
> >**Q8:** The example in Fig. 7 would be more informative if it showcased an example in which the sc-TMScore is increased by applying the proposed method when compared to baseline methods (even if it is a cherry-picked example).
>
> **Reply:** In the revision, we will update the figure to showcase an example where our method achieves a higher sc-TMScore compared to baseline methods. We note that such improvements are consistently observed across the majority of test cases, as reflected in our quantitative results.
>
>
>
> >**Q10:** For the reviewer, it is still not clear if the sequence in a state s\_t contains all the amino acid identities and these are mutated depending on the positions sampled, or if there are positions masked that are being filled. From Fig. 2(b), it seems that there are masked positions that are being filled, but it is not clear in the writing. Also, this affects the MCTS termination conditions, as I expected the MCTS to be terminated when the full sequence is generated (no need of the denoiser to calculate the reward function).
>
> **Reply:** In our framework, the sequence in a state s\_t is a partially generated sequence, which means there are positions masked that are being filled. And the MCTS terminated when the full sequence was generated and also the reward exceeded the predefined threshold. Importantly, to enable **intermediate feedback** during MCTS, we must evaluate partially generated sequences before reaching the final state. This is why we propose **a jumpy denoising strategy**, which completes the partially filled sequence via a fast approximation.
>
>
>
> > **Q12:** In the manuscript, the authors used ESM-3 as their inverse folding method, but any other protein inverse folding could be combined with the proposed MCTS and jumpy denoiser framework. Did the authors test other models as the inverse folding method?
>
> **Reply:** To the best of our knowledge, ESM-3 is the first structure-sequence unified model that supports arbitrary masking ratios for sequence completion, which makes it particularly suitable for our planning-based inverse folding framework. Other models, such as ProteinMPNN and KWDesign (which relies on ESM-2), do not support arbitrary masking ratios and typically require retraining or finetuning to enable structure-conditioned generation. This conflicts with several core design principles of ProtInvTree:
>
> 1. **Stepwise generation**: Our method incrementally constructs sequences, with each decision depending on the prior t−1 steps—requiring the model to flexibly accept partially completed sequences.
> 2. **Test-time reward-guided search**: ProtInvTree performs planning by querying the foundation model at test time, without retraining or gradient-based updates.
> 3. **Focus-and-grounding with jumpy denoising design**, which requires flexible infilling under arbitrary masking ratios.

---

> > ### Comment · Reviewer_z7LH · 2025-08-02
> > **Response to Rebuttal by Authors**
> >
> > Thank you for your detailed answers to my comments.
> >
> > The majority of my comments were effectively addressed. Still, I think it would be important to be reflected in the manuscript:
> >
> > 1. Add clarification on the jumpy denoiser, and also explain why ESM3 is the main model used.
> > 2. Evaluation metrics: Clarify which structure prediction network is used to calculate the designability metrics.
> > 3. Reproducibility: Make sure that the code is shared and the manuscript has all information for reproducibility. Details seem to be missing from the current version.
> >
> > I will reflect the rebuttal in my score, increasing it from 3 to 4 for my final justification.

---

> > > ### Author Response · Authors · 2025-08-02
> > >
> > > Thank you very much for your constructive feedback and for considering our rebuttal in your updated evaluation. We greatly appreciate your thoughtful comments and the opportunity to further improve the manuscript. We will revise the manuscript accordingly to address all of your suggestions.

---

> > > ### Author Response · Authors · 2025-08-06
> > > **Follow-up on Score Update**
> > >
> > > Thank you very much for your thoughtful feedback. We just wanted to kindly confirm whether the score has already been updated in the system, as we still see the initial score reflected. Thank you again for your time and support.

---

### Official Review · Reviewer_PyE2 · 2025-07-01

**Clarity:** 2
**Significance:** 2
**Originality:** 3
**Rating:** 4
**Confidence:** 3

**Summary:**

The paper proposes ProtInvTree, a training‑free inference framework that frames protein inverse folding as a reward‑guided Monte‑Carlo Tree Search (MCTS) problem. At each step, a residue position is first chosen to modify and then residues are sampled from a frozen protein language model (e.g., ESM‑3). Intermediate states are evaluated with a jumpy denoising shortcut that avoids full rollouts, using ESMFold to estimate a TM‑score‑based reward. The rewards are then backpropagated normally as in MCTS.

**Questions:**

- What is J(.) in Equation 10, and how does it save computation compared to standard decoding?
- What’s DDIM in line 214?
- Line 135: MPD → MDP

**Ethical Concerns:**

["NO or VERY MINOR ethics concerns only"]

**Final Justification:**

The authors conducted additional experiments to demonstrate the effectiveness of using MCTS decoding, which addressed my concerns. As a result, I have raised my rating to positive.

**Limitations:**

Yes, but I suggest the author also address the computational cost in the limitations section.

**Paper Formatting Concerns:**

I didn't notice anything.

**Quality:**

3

**Strengths And Weaknesses:**

- Strengths
    - The integration of MCTS into the inverse folding problem is intuitive and executed reasonably.
    - Results in Table 1-3 demonstrate the method’s effectiveness across multiple benchmarks, achieving SoTA performance.
    - Figure 3 shows the method’s effectiveness in balancing diversity and consistency.
- Weaknesses
    - Comparisons to other test-time scaling methods are lacking. Specifically, the author should compare MCTS to simpler methods like beam search and best-of-N to demonstrate the need for a more complex MCTS approach.
    - The explanations for the Jumpy Denoising strategy are unclear.
    - MCTS also brings additional computational costs, which should be noted and discussed.

---

> ### Author Rebuttal · Authors · 2025-07-30
>
> Thank you for your detailed and constructive feedback on our work. We appreciate your insights into both the strengths and weaknesses of our approach, and we address each of your points below:
> > **W1:** Comparisons to other test-time scaling methods are lacking. Specifically, the author should compare MCTS to simpler methods like beam search and best-of-N to demonstrate the need for a more complex MCTS approach.
>
> **Reply:** We compare our MCTS-based planner (ProtInvTree) against two commonly used test-time strategies: Beam Search and Best-of-N sampling. To ensure a fair comparison in computational cost, all methods are restricted to approximately the same number of model calls per design.
>
> | Models      | scTM-score (↑) | RMSD (↓) | Intermediate Feedback | Backtracking |
> | ----------- | -------------- | -------- | --------------------- | ------------ |
> | ESM-3       | 0.816          | 2.135    | ×                     | ×            |
> | Best-of-N   | 0.839          | 1.974    | ×                     | ×            |
> | Beam Search | 0.853          | 1.724    | √                     | ×            |
> | ProtInvTree | 0.881          | 1.513    | √                     | √            |
>
> Our experiments show that ProtInvTree consistently outperforms these baselines, which we attribute to its ability to leverage intermediate structural feedback and to perform both lookahead planning and backtracking-based revision.
>
> > **W2:** The explanations for the Jumpy Denoising strategy are unclear.
>
> **Reply:** We clarify that the Jumpy Denoising strategy is designed to efficiently estimate the final reward in MCTS without simulating full denoising trajectories. Specifically, instead of performing step-by-step denoising from noise to the final sequence, we adopt a single-step DDIM-based reverse sampling method that directly samples a denoised sequence Sample $\tilde{\mathbf{x_{T}}}$ from an intermediate state $\mathbf{x_{t+1}}$, conditioned on $\mathbf{c}$. Notably, the jumpy denoiser is not separately trained by us. It leverages the pretrained ESM-3 model, which is capable of filling in arbitrary mask ratios through its masked denoising objective.
>
>
>
> > **W3:**  MCTS also brings additional computational costs, which should be noted and discussed.
>
> **Reply:** Thank you for raising this important concern. We agree that test-time computational efficiency is critical, particularly for methods that involve large pretrained models and structure prediction. We provide a detailed analysis of the compute–performance trade-off. Specifically, for each given backbone, we generate 10 designs and compute the average generation time for each design. We fix the depth and scale the expand number from 2 to 6, as follows:
>
> | Expand Number                      |   2 |   3 |   4 |   5 |   6 |
> | --------------------------- | ----: | ----: | ----: | ----: | ----: |
> | sc-TMScore                  | 0.859 | 0.865 | 0.868 | 0.872 | 0.878 |
> | Inference Time / design (s) |  6.72 | 11.43 | 21.20 | 32.29 | 43.92 |
>
> We also scaled the number of sampling iterations for ESM-3 to assess its performance–efficiency trade-off. The resulting sc-TMScore and average inference time per design are summarized in the table below:
>
> | Iteration Number            |     5 |    10 |    50 |    80 |   100 |
> | --------------------------- | ----: | ----: | ----: | ----: | ----: |
> | sc-TMScore                  | 0.816 | 0.822 | 0.825 | 0.824 | 0.826 |
> | Inference Time / design (s) | 0.438 | 0.706 | 3.397 | 5.481 | 6.897 |
>
> We also evaluate the method of Best of N, which generates 10 designs and finds the best one with 58.19, with an average sc-TMScore of 0.839.
>
>
>
>
> > **Q1:** What is J(.) in Equation 10, and how does it save computation compared to standard decoding?
>
> **Reply:** J(·) approximates the single-step reverse denoising $p(\mathbf{x_T} \mid \mathbf{x_{t+1}},\mathbf{c})$, This contrasts with standard diffusion decoding $\prod_{s=t+1}^{T-1} p_{\theta}(\mathbf{x}_{s+1} \mid \mathbf{x}_s, \mathbf{c})$, which requires iterative denoising steps. We bypass this iterative process and generate the final sample in one step, significantly reducing the computational cost for each node evaluation in the MCTS procedure.
>
>
>
> > **Q2:** What’s DDIM in line 214?
>
> **Reply:** DDIM refers to Denoising Diffusion Implicit Models [1], a sampling method used in diffusion models that accelerates the generation by jumpy denoising. In our method, we apply a single-step DDIM sampling to approximate the final denoised sequence, substantially reducing computational cost during MCTS rollout. We will add the citation [1] in the final version for clarity.
>
>
>
> > **Q3:** Line 135: MPD → MDP
>
> **Reply:** Thank you for pointing this out. We will correct the typo in the revised version of the manuscript.
>
> > **Q4:** Address the computational cost in the limitations section.
>
> **Reply:** Yes, we discuss the computational cost in reply of W3, and we will add the limitations in our revised version.
>
> [1] Jiaming Song et al, Denoising Diffusion Implicit Models, ICLR 2021

---

> > ### Comment · Reviewer_PyE2 · 2025-08-04
> >
> > Thank you for the additional experiments. These results addressed my concerns. I will raise my score from 3 to 4.

---

> > > ### Author Response · Authors · 2025-08-06
> > > **Follow-up on Score Update**
> > >
> > > Thank you very much for your thoughtful feedback and for acknowledging that our additional experiments have addressed your concerns. We truly appreciate your consideration and your comment indicating that you would raise the score from 3 to 4.
> > > We just wanted to kindly confirm whether the score has already been updated in the system, as we still see the initial score reflected. Thank you again for your support and for helping us improve our manuscript.

---

### Official Review · Reviewer_mTNz · 2025-07-02

**Clarity:** 3
**Significance:** 3
**Originality:** 3
**Rating:** 5
**Confidence:** 4

**Summary:**

This paper presents ProtInvTree, a reward-guided tree search framework for protein inverse folding. It reformulates sequence generation as a step-wise decision process using Monte Carlo Tree Search, enabling exploration of diverse sequences consistent with a target structure. Key innovations include a two-stage focus-and-grounding mechanism for position and residue selection, and a jumpy denoising strategy for efficient evaluation. Built on pretrained models without retraining, ProtInvTree achieves state-of-the-art performance in terms of structural consistency, diversity, and novelty. Experimental validation is pending.

**Questions:**

A)
1. **Intermediate Feedback and Uncertainty**: Section 4.1 emphasizes the benefits of incorporating intermediate feedback, tracking uncertainty, and enabling backtracking via tree-based search. Could you provide empirical evidence to isolate these contributions? For example, how much do they improve performance over simpler strategies?

2. **Beam Search Baseline**: Have you considered comparing ProtInvTree to a **beam search baseline** that does not include backtracking or uncertainty estimation? Including such a baseline would clarify whether the proposed advantages of tree search are essential or incremental.

3. **Ablation of Planning Features**: Can you ablate specific planning features—e.g., removing backtracking, removing jumpy denoising, or using greedy selection—to assess their individual impact on structural consistency, diversity, and novelty?

4. **Uncertainty Quantification**: How is uncertainty explicitly tracked and used during search in your implementation? If applicable, showing how this improves exploration or stability would help justify its inclusion.



B)
1. **Novelty Collapse Analysis**: Have you evaluated how novelty changes alongside diversity as planning depth and expansion width increase (e.g., in Figure 5)? Including novelty plots would clarify whether deeper search also leads to convergence toward native sequences.

2. **Failure Mode Characterization**: It would strengthen the work to systematically identify conditions under which ProtInvTree collapses to native-like sequences. Could you analyze thresholds (e.g., depth ≥ X or expansion ≥ Y) beyond which the method fails to explore novel solutions?

3. **Parameter Sensitivity Study**: Consider conducting a controlled sweep over planning parameters to map the diversity–novelty–consistency landscape. This would help practitioners tune the method for different design goals (e.g., conservative vs. exploratory designs).

4. **Recovery-Oriented Trade-off**: Would adding an explicit recovery-based reward term bias the search toward the native sequence? A comparison with and without such terms could reveal how tightly ProtInvTree adheres to or diverges from native recovery.

5. **Structure Complexity Effects**: Does the convergence behavior vary across protein types (e.g., different lengths, fold classes, or disorder levels)? Reporting such breakdowns might uncover structure-specific tuning needs.

**Ethical Concerns:**

["NO or VERY MINOR ethics concerns only"]

**Final Justification:**

I maintain my score of **5: Accept**, as the authors have adequately addressed key concerns raised in the initial review and demonstrated the technical strength and potential impact of the work.

**Resolved Issues:**

* **Beam Search and Simpler Baselines:** The authors added comparisons against both Beam Search and Best-of-N baselines under matched compute budgets. This helps contextualize the gains from MCTS, particularly with respect to structural consistency and novelty.

* **Planning Feature Ablations:** A thorough ablation study was provided, showing the individual contributions of intermediate feedback, uncertainty-driven exploration, and backtracking. This validates the benefit of each planning component.

* **Novelty Collapse and Scaling Behavior:** The authors added novelty plots and extended the planning depth/width experiments. Results indicate a gradual, not abrupt, decline in novelty, and no evidence of full collapse to native-like sequences.

* **Test-Time Compute Trade-off:** The authors now report detailed timing vs. accuracy data, which quantifies the inference cost of deeper planning.

* **Uncertainty Quantification:** The reply clarifies that uncertainty is handled via the standard UCT mechanism. Ablation of this term showed a measurable drop in performance, supporting its utility.

* **Structure-Specific Analysis:** The authors provided additional breakdowns across fold classes and protein lengths, revealing meaningful differences in performance and novelty trends.

**Partially Addressed or Remaining Issues:**

* **Functional Validation:** As noted, experimental validation is out of scope, and while structural fidelity is a strong proxy, it limits confidence in real-world utility. This is common in protein design papers but remains a caveat.

* **Recovery-Biased Rewards:** The authors rightly note that adding recovery-based rewards would leak ground truth. However, an approximate alternative (e.g., homology-guided regularization) could be explored in future work to assess biases in the method.

**Limitations:**

Yes

**Quality:**

3

**Strengths And Weaknesses:**

### **Strengths**

**Quality:**
The paper proposes ProtInvTree, a reward-guided MCTS framework for protein inverse folding that operates entirely at test time. Built on top of pretrained models (ESM-3), it delivers strong performance across benchmarks (CATH 4.2/4.3, TS45) without additional training. The method achieves state-of-the-art structural consistency while improving sequence diversity and novelty. Jumpy denoising offers an efficient approximation for reward evaluation.

**Clarity:**
The paper is clearly written, with intuitive diagrams and a detailed description of the method.

**Significance:**
Test-time compute scaling is a key strength. The method enables more deliberate search at inference without retraining.

**Originality:**
Applying MCTS to protein sequence design is novel. The focus-and-grounding mechanism and jumpy denoising strategy are also original contributions.


### **Weaknesses**

* The claimed benefits of planning (intermediate feedback, uncertainty tracking, backtracking) are not directly validated. There is no comparison to beam search or other simpler baselines that lack these features.
* Figure 5 shows a drop in diversity with deeper planning, but the paper does not analyze how **novelty** changes under the same conditions. This limits insight into when and how the method converges toward native-like sequences.
* The test-time compute–performance trade-off is not quantified in practical terms (e.g., runtime vs. accuracy).
* There is no experimental or functional validation of designed sequences.

---

> ### Author Rebuttal · Authors · 2025-07-30
>
> Thank you for your thoughtful and detailed feedback. We appreciate your positive assessment of the strengths of our work, as well as your suggestions for areas of improvement. We address each of your comments in detail below:
> > **W1 & Q1 & Q2:** The claimed benefits of planning (intermediate feedback, uncertainty tracking, backtracking) are not directly validated. There is no comparison to beam search or other simpler baselines that lack these features. Add Beam Search Baseline.
>
> **Reply:** We compare our MCTS-based planner (ProtInvTree) against two commonly used test-time strategies: Beam Search and Best-of-N sampling. To ensure a fair comparison in computational cost, all methods are restricted to approximately the same number of model calls per design.
>
> | Models        | scTM-score (↑) | RMSD (↓) | Intermediate Feedback | uncertainty tracking | Backtracking |
> |---|---|---|---|---|---|
> | ESM-3         | 0.816          | 2.135    | ×                     | ×                   | ×            |
> | Best-of-N     | 0.839          | 1.974    | ×                     | ×                   | ×            |
> | Beam Search   | 0.853          | 1.724    | √                     | ×                   | ×            |
> | ProtInvTree   | 0.881          | 1.513    | √                     | √                   | √            |
>
>
>
> > **W2 & Q5:** Novelty & Collapse “Figure 5 shows diversity decreasing with deeper planning, but novelty is not reported. How does novelty behave and when does the search collapse to native‑like sequences?”
>
> **Reply:** We will add novelty curves in the revised Fig. 5. The key trend is a smooth trade‑off—better scTM comes with a gradual, not abrupt, drop in novelty. Even at our most aggressive settings the average fraction of mutated residues remains > 49 %. A condensed view of the sweep is below:
>
> | Metric                     | N=2   | N=3   | N=4   | N=5   | N=6   |
> |----------------------------|------:|------:|------:|------:|------:|
> | sc-TMScore                 | 0.859 | 0.865 | 0.868 | 0.872 | 0.878 |
> | Novelty                    | 0.556 | 0.534 | 0.517 | 0.503 | 0.492 |
>
> | Metric    | D=2   | D=3   | D=4   | D=5   | D=6   | D=7   |
> |--------|------:|------:|------:|------:|------:|------:|
> | sc-TMScore | 0.838 | 0.849 | 0.852 | 0.859 | 0.858 | 0.859 |
> | Novelty  | 0.598 | 0.577 | 0.561 | 0.556 | 0.545 | 0.528 |
>
> > **W3:** “Quantify the test‑time compute vs. accuracy trade‑off.”
>
>  **Reply:** We provide a detailed analysis of the compute–performance trade-off. Specifically, for each given backbone, we generate 10 designs and compute the average generation time for each design. We fix the depth and scale the expand number from 2 to 6, as follows:
>
> | Metric                     | N=2   | N=3   | N=4   | N=5   | N=6   |
> |----------------------------|------:|------:|------:|------:|------:|
> | sc-TMScore                 | 0.859 | 0.865 | 0.868 | 0.872 | 0.878 |
> | Inference Time / design (s)| 6.72  | 11.43 | 21.20 | 32.29 | 43.92 |
>
> > **W4:** There is no experimental or functional validation
>
>  **Reply:** Wet‑lab assays are beyond scope. In protein engineering, function is overwhelmingly constrained by 3‑D structure—hence the maxim “structure dictates function.” A design that reproduces the target fold with higher scTMScore is therefore more likely to preserve (or enable) the desired biochemical activity.
>
> > **Q1 & Q2 & Q3**: Intermediate Feedback & Uncertainty. Ablation of Planning Features.
>
>  **Reply:**  Starting from the full ProtInvTree, we conducted an ablation study by disabling each key mechanism in isolation to assess its contribution:
>
> | Variant        | Intermed. feedback | Uncertainty (UCT term) | Backtracking |  scTMscore (↑) | Novelty (↑) | Diversity (↑) |
> | ----------------------------- | ------------------ | ---------------------- | ------------- | ---------------: | --------: | ----------: |
> | **Full ProtInvTree**          | ✓                  | ✓                      | ✓             |        0.881 |       0.498 |   0.159 |
> | – backtracking (no path revision) | ✓                  | ✓                     | ✗            | 0.865 |       0.479 |         0.138 |
> | – UCT (greedy selection) | ✓                  | ✗                     | ✓            |         0.874 |       0.461 |         0.127 |
> | Beam Search | ✓                 | ✗                      | ✗             |       0.853 | 0.457 | 0.118 |
> | ESM‑3 (no search)      | ✗                  | ✗                      | ✗             |            0.816 | 0.449 | 0.107 |
>
> We also evaluated the impact of jumpy denoising on runtime performance. Specifically, for each given backbone, we generate 10 designs and compute the average generation time for each design.
>
> | Variant           | N = 2 | N = 3 | N = 4 | N = 5 | N = 6 |
> | ----------------- | ----: | ----: | ----: | ----: | ----: |
> | ProtInvTree       |  6.72 | 11.43 | 21.20 | 32.29 | 43.92 |
> | – jumpy denoising |  6.78 | 11.54 | 21.38 | 32.61 | 44.66 |
>
>
>
> > **Q4:** Uncertainty Quantification
>
>  **Reply:** ProtInvTree follows the classic UCT rule (Eq. 1 in the paper):
>  $\text{UCT}(s)=V(s)+\omega\sqrt{\frac{\ln N(\text{parent}(s))}{N(s)}} .$
>  We treat this visit‑count–based bonus as an implicit measure of epistemic uncertainty—lower $N(s)$ ≈ higher uncertainty. No additional statistics are stored or tuned. We provice more results by setting $w\!=\!0$ (greedy value search) as follows:
>
> | Variant                                      | scTM‑score (↑) | RMSD (↓)  | Novelty (↑) |
> | -------- | ----- | ---- | ----- |
> | ProtInvTree (full, $\omega=0.01$)             | **0.881**      | **1.513** | **0.498**   |
> | ProtInvTree w/o uncertainty term ($\omega=0$) | 0.874          | 1.562     | 0.461       |
>
>  Removing the uncertainty‑driven exploration degrades structural quality and shrinks the explored sequence space, demonstrating that the built‑in visit‑count heuristic already captures useful uncertainty information.
>
>
>
> > **Q6:**  Failure Mode Characterization
>
> **Reply:** We attempted to explore higher values (as shown in the following) as further extreme cases.
>
> | Expansion Number (N) | Planning Depth (D) | Novelty (↑) |
> | ---- | ------ | ----- |
> | 10 | 10 | 0.485       |
> | 20 | 10  | 0.482       |
> | 20 | 20                 | 0.476       |
> | 10 | 50                 | 0.475       |
>
> However, we regret that when both the expansion number and planning depth exceed this level, the memory consumption of the model exceeds the capacity of a single A100 GPU (80 GB). Despite this, our current scaling experiments already reveal a clear saturation point: although novelty decreases slightly as both expansion number and planning depth increase, the decline is gradual and does not indicate a full collapse to native-like sequences.
>
>
> > **Q7:** Parameter Sensitivity Study
>
>  **Reply:** To systematically characterize the diversity–novelty–consistency landscape, beyond the scaling analysis we have already conducted on planning depth and expansion number, we conducted a controlled sweep over several key planning hyperparameters: the UCT exploration weight $\omega$, the sampling temperature $T$, the reward threshold used for termination, and selection number schedules.
>
> | $\omega$ | Consistency (↑) | Novelty (↑) | Diversity (↑) |
> | --- | --- | --- | --- |
> | 0.01 | 0.881 | 0.498 | 0.159 |
> | 0.05 | 0.871 | 0.519 | 0.175 |
> | 0.10 | 0.860 | 0.538 | 0.194 |
>
> | $T$ | Consistency (↑) | Novelty (↑) | Diversity (↑) |
> | --- | --- | --- | --- |
> | 0.1 | 0.882 | 0.468 | 0.109 |
> | 0.3 | 0.881 | 0.498 | 0.159 |
> | 0.5 | 0.856 | 0.564 | 0.378 |
> | 0.7 | 0.824 | 0.611 | 0.607 |
>
> | Threshold | Consistency (↑) | Novelty (↑) | Diversity (↑) |
> | --- | --- | --- | --- |
> | 0.0 | 0.859 | 0.556 | 0.224 |
> | 0.7 | 0.881 | 0.498 | 0.159 |
> | 0.9 | 0.887 | 0.496 | 0.146 |
>
> | Schedule | Consistency (↑) | Novelty (↑) | Diversity (↑) |
> | --- | --- | --- | --- |
> | cosine | 0.881 | 0.498 | 0.159 |
> | linear | 0.879 | 0.474 | 0.127 |
>
> > **Q8:**  Recovery-Oriented Trade-off
>
>  **Reply:** We agree that introducing an explicit recovery-based reward would bias the search toward the native target.  But we do not perform this ablation because adding a recovery term would require revealing the ground‑truth sequence during generation.
>
> > **Q9:** Structure Complexity Effects.
>
>  **Reply:** We thank the reviewer for this valuable question. As partially reflected in Table 1, our evaluation already includes a breakdown of model performance across different protein lengths (i.e., "Short" proteins with length ≤ 100, "Single-chain" proteins, and "All" proteins).
>
> The fold classes results are shown below:
>
> | Fold Class   | scTMscore (↑) | RMSD(↓)| Novelty (↑) |
> | ----------------- |  ------------: |------: | ----------: |
> | Mainly Alpha      | 0.855 | 1.548 | 0.616 |
> | Mainly Beta       | 0.830 | 1.710 | 0.503 |
> | Alpha-Beta    | 0.908 | 1.488 |0.448 |
> | Few Secondary | 0.764 | 1.349 | 0.553 |

---

### Public Comment · ~Tao_Huang18 · 2025-12-12
**No code was provided.**

The code is available at https://github.com/A4Bio/ProteinInvBench/.
I couldn’t find the code in this GitHub repository.

---

### Public Comment · ~Jiahao_Li7 · 2026-02-27
**No code was provided in Github**

The code was provided in the link, https://github.com/A4Bio/ProteinInvBench/, as described in abstract, but I fail to find it in Github.

---

### Note · Authors · 2025-08-12

We express our gratitude to all the reviewers for their valuable insights! We appreciate all of you for your comments highlighting the strengths of our work for a summary.

- **Originality (Reviewer mTNz, PyE2, z7LH, mMYi)**: Applying MCTS to protein sequence design is novel and well-executed. The focus-and-grounding and jumpy denoising strategies are also original contributions.
- **Impressive performance across diverse metrics (Reviewer mTNz, PyE2, z7LH, mMYi)**: The method outperforms SOTA inverse folding models. The benchmarking is comprehensive, extending beyond recovery to evaluate Pareto-optimal trade-offs among structural consistency, novelty, and diversity.
- **Significance (Reviewer mTNz, z7LH)**: Test-time compute scaling is a key strength, enabling more deliberate search at inference without retraining.
- **Well-clarified (Reviewer mTNz, mMYi)**: The paper is well-written and well-structured, with intuitive diagrams and detailed descriptions, making it a pleasure to read and easy to follow.

A major concern is the discussion of computational costs. We address this with a detailed compute–performance trade-off analysis, generating 10 designs per backbone and reporting average generation time and sc-TMScore. We also vary ESM-3’s sampling iterations for comparison. Another concern is the lack of direct validation for planning benefits. We address this by comparing variants without backtracking or UCT, and baselines such as Beam Search, Best-of-N, and no search.

We are trying to address the concerns and polish our paper in the revised version. Specifically, we will:

- Add a detailed discussion of computational costs, ablation studies, and hyperparameter configurations.
- Clarify the jumpy denoiser, the structure prediction network used for designability metrics, and the rationale for using ESM-3.
- Release the code and ensure all reproducibility details are included.
- Revise statements to eliminate spelling errors and ambiguities.

Finally, we would like to express our sincere appreciation and excitement that several reviewers have recognized the value of our work. After the discussion phase, the final scores reached **5, 5, 4, and 4**. We believe our study not only **establishes new evaluation standards for protein inverse folding** but also introduces a novel **test-time scaling paradigm**, providing promising directions for further exploration. We believe that it is worth publishing to stimulate further discussion.

---

### Decision · Program_Chairs · 2025-09-17

**Decision:**

Accept (spotlight)

**Comment:**

After the review, rebuttal and discussion all reviewers recommend acceptance. The AC agrees.
The reviewers appreciated the novelty (MCTS) and found the paper well executed. Most questions have been resolved satisfactory.